# Deep Learning Games

**Dale Schuurmans***
Google
daes@ualberta.ca

**Martin Zinkevich**
Google
martinz@google.com

## Abstract

We investigate a reduction of supervised learning to game playing that reveals new connections and learning methods. For convex one-layer problems, we demonstrate an equivalence between global minimizers of the training problem and Nash equilibria in a simple game. We then show how the game can be extended to general acyclic neural networks with differentiable convex gates, establishing a bijection between the Nash equilibria and critical (or KKT) points of the deep learning problem. Based on these connections we investigate alternative learning methods, and find that regret matching can achieve competitive training performance while producing sparser models than current deep learning strategies.

## 1 Introduction

In this paper, we investigate a new approach to reducing supervised learning to game playing. Unlike well known reductions [8, 29, 30], we avoid duality as a necessary component in the reduction, which allows a more flexible perspective that can be extended to deep models. An interesting finding is that the no-regret strategies used to solve large-scale games [35] provide effective stochastic training methods for supervised learning problems. In particular, regret matching [12], a step-size free algorithm, appears capable of efficient stochastic optimization performance in practice.

A central contribution of this paper is to demonstrate how supervised learning of a directed acyclic neural network with differentiable convex gates can be expressed as a simultaneous move game with simple player actions and utilities. For variations of the learning problem (i.e. whether regularization is considered) we establish connections between the critical points (or KKT points) and Nash equilibria in the corresponding game. As expected, deep learning games are not simple, since even approximately training deep models is hard in the worst case [13]. Nevertheless, the reduction reveals new possibilities for training deep models that have not been previously considered. In particular, we discover that regret matching with simple initialization can offer competitive training performance compared to state-of-the-art deep learning heuristics while providing sparser solutions.

Recently, we have become aware of unpublished work [2] that also proposes a reduction of supervised deep learning to game playing. Although the reduction presented in this paper was developed independently, we acknowledge that others have also begun to consider the connection between deep learning and game theory. We compare these two specific reductions in Appendix J, and outline the distinct advantages of the approach developed in this paper.

## 2 One-Layer Learning Games

We start by considering the simpler one-layer case, which allows us to introduce the key concepts that will then be extended to deep models. Consider the standard supervised learning problem where one is given a set of paired data $\{(x_t, y_t)\}_{t=1}^T$, such that $(x_t, y_t) \in \mathcal{X} \times \mathcal{Y}$, and wishes to learn a

predictor $h : \mathcal{X} \to \mathcal{Y}$. For simplicity, we assume $\mathcal{X} = \mathbb{R}^m$ and $\mathcal{Y} = \mathbb{R}^n$. A standard generalized linear model can be expressed as $h(x) = \phi(\theta x)$ for some output transfer function $\phi : \mathbb{R}^n \to \mathbb{R}^n$ and matrix $\theta \in \mathbb{R}^{n \times m}$ denoting the trainable parameters of the model. Despite the presence of the transfer function $\phi$, such models are typically trained by minimizing an objective that is convex in $z = \theta x$.

**OLP (One-layer Learning Problem)** Given a loss function $\ell : \mathbb{R}^n \times \mathbb{R}^n \to \mathbb{R}$ that is convex in the first argument, let $\ell_t(z) = \ell(z, y_t)$ and $L_t(\theta) = \ell_t(\theta x_t)$. The training problem is to minimize $L(\theta) = T^{-1} \sum_{t=1}^{T} L_t(\theta)$ with respect to the parameters $\theta$.

We first identify a simple game whose Nash equilibria correspond to global minima of the one-layer learning problem. This basic relationship establishes a connection between supervised learning and game playing that we will exploit below. Although this reduction is not a significant contribution by itself, the one-layer case allows us to introduce some key concepts that we will deploy later when considering deep neural networks. A one-shot **simultaneous move game** is defined by specifying: a set of players, a set of actions for each player, and a set of utility functions that specify the value to each player given a joint action selection [36, Page 9] (also see Appendix E). Corresponding to the OLP specified above, we propose the following game.

**OLG (One-layer Learning Game)** There are two players, a protagonist $p$ and an antagonist $a$. The protagonist chooses a parameter matrix $\theta \in \mathbb{R}^{m \times n}$. The antagonist chooses a set of $T$ vectors and scalars $\{a_t, b_t\}_{t=1}^{T}$, $a_t \in \mathbb{R}^n$, $b_t \in \mathbb{R}$, such that $a_t^\top z + b_t \leq \ell_t(z)$ for all $z \in \mathbb{R}^n$; that is, the antagonist chooses an *affine minorant* of the local loss for each training example. Both players make their action choice without knowledge of the other player's choice. Given a joint action selection $(\theta, \{a_t, b_t\})$ we define the utility of the antagonist as $U^a = T^{-1} \sum_{t=1}^{T} a_t^\top \theta x_t + b_t$, and the utility of the protagonist as $U^p = -U^a$. This is a two-person zero-sum game with continuous actions.

A **Nash equilibrium** is defined by a joint assignment of actions such that no player has any incentive to deviate. That is, if $\sigma^p = \theta$ denotes the action choice for the protagonist and $\sigma^a = \{a_t, b_t\}$ the choice for the antagonist, then the joint action $\sigma = (\sigma^p, \sigma^a)$ is a Nash equilibrium if $U^p(\tilde{\sigma}^p, \sigma^a) \leq U^p(\sigma^p, \sigma^a)$ for all $\tilde{\sigma}^p$, and $U^a(\sigma^p, \tilde{\sigma}^a) \leq U^a(\sigma^p, \sigma^a)$ for all $\tilde{\sigma}^a$.

Using this characterization one can then determine a bijection between the Nash equilibria of the OLG and the global minimizers of the OLP.

**Theorem 1** *(1) If $(\theta^*, \{a_t, b_t\})$ is a Nash equilibrium of the OLG, then $\theta^*$ must be a global minimum of the OLP. (2) If $\theta^*$ is a global minimizer of the OLP, then there exists an antagonist strategy $\{a_t, b_t\}$ such that $(\theta^*, \{a_t, b_t\})$ is a Nash equilibrium of the OLG.* **(All proofs are given in the appendix.)**

Thus far, we have ignored the fact that it is important to control model complexity to improve generalization, not merely minimize the loss. Although model complexity is normally controlled by regularizing $\theta$, we will find it more convenient to equivalently introduce a constraint $\theta \in \Theta$ for some convex set $\Theta$ (which we assume satisfies an appropriate constraint qualification; see Appendix C). The learning problem and corresponding game can then be modified accordingly while still preserving the bijection between their solution concepts.

**OCP (One-layer Constrained Learning Problem)** Add optimization constraint $\theta \in \Theta$ to the OLP.
**OCG (One-layer Constrained Learning Game)** Add protagonist action constraint $\theta \in \Theta$ to OLG.

**Theorem 2** *(1) If $(\theta^*, \{a_t, b_t\})$ is a Nash equilibrium of the OCG, then $\theta^*$ must be a constrained global minimum of the OCP. (2) If $\theta^*$ is a constrained global minimizer of the OCP, then there exists an antagonist strategy $\{a_t, b_t\}$ such that $(\theta^*, \{a_t, b_t\})$ is a Nash equilibrium of the OCG.*

## 2.1 Learning Algorithms

The tight connection between convex learning and two-person zero-sum games raises the question of whether techniques for finding Nash equilibria might offer alternative training approaches. Surprisingly, the answer appears to be yes.

There has been substantial progress in on-line algorithms for finding Nash equilibria, both in theory [5, 24, 34] and practice [35]. In the two-person zero-sum case, large games are solved by pitting two regret-minimizing learning algorithms against each other, exploiting the fact that when both achieve a regret rate of $\epsilon/2$, their respective average strategies form an $\epsilon$-Nash equilibrium [38]. For the game as described above, where the protagonist action is $\theta \in \Theta$ and the antagonist action is denoted $\sigma_a$,

we imagine playing in rounds, where on round $k$ the joint action is denoted by $\sigma^{(k)} = (\theta^{(k)}, \sigma_a^{(k)})$. Since the utility function for each player $U^i$ for $i \in \{p, a\}$, is affine in their own action choice for any fixed action chosen by the other player, each faces an online convex optimization problem [37] (note that maximizing $U^i$ is equivalent to minimizing $-U^i$; see also Appendix G). The **total regret** of a player, say the protagonist, is defined with respect to their utility function after $K$ rounds as $R^p(\sigma^{(1)} \ldots \sigma^{(K)}) = \max_{\theta \in \Theta} \sum_{k=1}^{K} U^p(\theta, \sigma_a^{(k)}) - U^p(\theta^{(k)}, \sigma_a^{(k)})$. (Nature can also be introduced to choose a random training example on each round, which simply requires the definition of regret to be expressed in terms of expectations over nature's choices.)

To accommodate regularization in the learning problem, we impose parameter constraints $\Theta$. A particularly interesting case occurs when one defines $\Theta = \{\theta : \|\theta\|_1 \leq \beta\}$, since the $L_1$ ball constraint is equivalent to imposing $L_1$ regularization. There are two distinct advantages to $L_1$ regularization in this context. First, as is well known, $L_1$ encourages sparsity in the solution. Second, and much less appreciated, is the fact that *any* polytope constraint allows one to reduce the constrained online convex optimization problem to learning from expert advice over a finite number of experts [37]: Given a polytope $\Theta$, define the **convex hull basis** $H(\Theta)$ to be a matrix whose columns are the vertices in $\Theta$. An expert can then be assigned to each vertex in $H(\Theta)$, and an algorithm for learning from expert advice can then be applied by mapping its strategy on round $k$, $\rho^{(k)}$ (a probability distribution over the experts), back to an action choice in the original problem via $\theta^{(k)} = H(\Theta)\rho^{(k)}$, while the utility vector on round $k$, $u^{(k)}$, can be passed back to the experts via $H(\Theta)^\top u^{(k)}$ [37].

Since this reduction allows any method for learning from expert advice to be applied to $L_1$ constrained online convex optimization, we investigated whether alternative algorithms for supervised training might be uncovered. We considered two algorithms for learning from expert advice: the normalized **exponentiated weight algorithm** (EWA) [22, 32] (Algorithm 3); and **regret matching** (RM), a simpler method from the economics and game theory literature [12] (Algorithm 2). For supervised learning, these algorithms operate by using a stochastic sample of the gradient to perform their updates (outer loop Algorithm 1). EWA possesses superior regret bounds that demonstrate only a logarithmic dependence on the number of actions; however RM is simpler, hyperparameter-free, and still possesses reasonable regret bounds [9, 10]. Although exponentiated gradient methods have been applied to supervised learning [18, 32], we not aware of any previous attempt to apply regret matching to supervised training. We compared these to **projected stochastic gradient descent** (PSGD), which is the obvious modification of stochastic gradient descent (SGD) that retains a similar regret bound [7, 28] (Algorithm 4).

## 2.2 Evaluation

To investigate the utility of these methods for supervised learning, we conducted experiments on synthetic data and on the MNIST data set [20]. Note that PSGD and EWA have a step size parameter, $\eta^{(k)}$, that greatly affects their performance. The best regret bounds are achieved for step sizes of the form $\eta k^{-1/2}$ and $\eta \log(m) k^{-1/2}$ respectively [28]; we also tuned $\eta$ to generate the best empirical results. Since the underlying optimization problems are convex, these experiments merely focus on the speed of convergence to a global minimum of the constrained training problem.

The first set of experiments considered synthetic problems. The data dimension was set to $m = 10$, and $T = 100$ training points were drawn from a standard multivariate Gaussian. For univariate prediction, a random hyperplane was chosen to label the data (hence the data was linearly separable, but not with a large margin). The *logistic* training loss achieved by the running average of the protagonist strategy $\bar{\theta}$ over the entire training set is plotted in Figure 1a. For multivariate prediction, a $4 \times 10$ target matrix, $\theta^*$, was randomly generated to label training data by $\arg\max(\theta^* x_t)$. The training *softmax* loss achieved by the running average of the protagonist strategy $\bar{\theta}$ over the entire training set is shown in Figure 1b. The third experiment was conducted on MNIST, which is an $n = 10$ class problem over $m = 784$ dimensional inputs with $T = 60,000$ training examples, evidently not linearly separable. For this experiment, we used mini-batches of size 100. The training loss of the running average protagonist strategy $\bar{\theta}$ (single run) is shown in Figure 1c. The apparent effectiveness of RM in these experiments is a surprising outcome. Even after tuning $\eta$ for both PSGD and EWA, they do not surpass the performance of RM, which is hyperparameter free. We did not anticipate this observation; the effectiveness of RM for supervised learning appears not to have been previously noticed. (We do not expect RM to be competitive in high dimensional *sparse* problems, since its regret bound has a square root and not a logarithmic dependence on $n$ [9].)

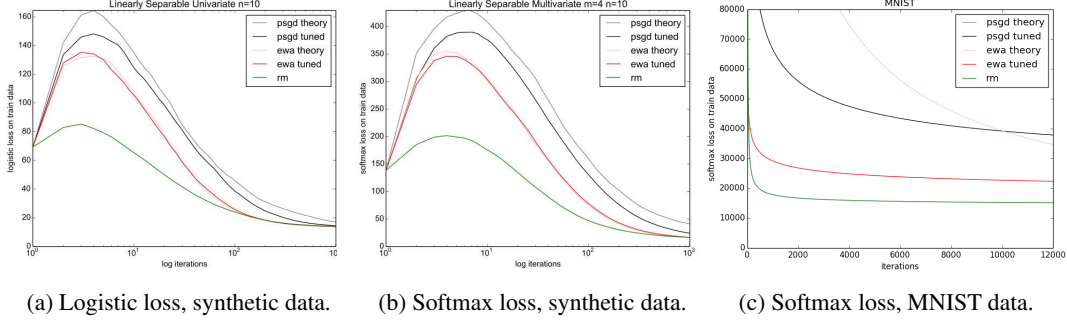

| (a) Logistic loss, synthetic data. | (b) Softmax loss, synthetic data. | (c) Softmax loss, MNIST data. |

Figure 1: Training loss achieved by different no-regret algorithms. Subfigures (a) and (b) are averaged over 100 repeats, log scale x-axis. Subfigure (c) is averaged over 10 repeats (psgd theory off scale).

## 3 Deep Learning Games

A key contribution of this paper is to show how the problem of training a feedforward neural network with differentiable convex gates can be reduced to a game. A practical consequence of this reduction is that it suggests new approaches to training deep models that are inspired by methods that have recently proved successful for solving massive-scale games.

**Feedforward Neural Network**    A feedforward neural network is defined by a directed acyclic graph with additional objects attached to the vertices and edges. The network architecture is specified by $N = (V, E, I, O, F)$, where $V$ is a set of vertices, $E \subseteq V \times V$ is a set of edges, $I = \{i_1 \ldots i_m\} \subset V$ is a set of input vertices, $O = \{o_1 \ldots o_n\} \subset V$ is a set of output vertices, and $F = \{f_v : v \in V\}$ is a set of activation functions, where $f_v : \mathbb{R} \to \mathbb{R}$. The trainable parameters are given by $\theta : E \to \mathbb{R}$.

In the graph defined by $G = (V, E)$, a **path** $(v_1, ..., v_k)$ consists of a sequence of vertices such that $(v_j, v_{j+1}) \in E$ for all $j$. A **cycle** is a path where the first and last vertex are equal. We assume that $G$ contains no cycles, the input vertices have no incoming edges (i.e. $(u, i) \notin E$ for all $i \in I$, $u \in V$), and the output vertices have no outgoing edges (i.e. $(o, v) \notin E$ for all $o \in O$, $v \in V$). A directed acyclic graph generates a partial order $\leq$ on the vertices where $u \leq v$ if and only if there is a path from $u$ to $v$. For all $v \in V$, define $E_v = \{(u, u') \in E : u' = v\}$. The network is related to the training data by assuming $|I| = m$, the number of input vertices corresponds to the number of input features, and $|O| = n$, the number of output vertices corresponds to the number of output dimensions. It is a good idea (but not required) to have two additional bias inputs, whose corresponding input features are always set to $0$ and $1$, respectively, and have edges to all non-input nodes in the graph. Usually, the activation functions on input and output nodes are the identity, i.e. $f_v(x) = x$ for $v \in I \cup O$.

Given a training input $x_t \in \mathbb{R}^m$, the computation of the network $N$ is expressed by a circuit value function $c_t$ that assigns values to each vertex based on the partial order over vertices:

$$c_t(i_k, \theta) = f_{i_k}(x_{tk}) \text{ for } i_k \in I; \quad c_t(v, \theta) = f_v\big(\textstyle\sum_{u:(u,v) \in E} c_t(u, \theta)\theta(u, v)\big) \text{ for } v \in V - I. \quad (1)$$

Let $c_t(\mathbf{o}, \theta)$ denote the vector of values at the output vertices, i.e. $(c_t(\mathbf{o}, \theta))_k = c_t(o_k, \theta)$. Since each $f_v$ is assumed differentiable, the output $c_t(\mathbf{o}, \theta)$ must also be differentiable with respect to $\theta$.

When we wish to impose constraints on $\theta$ we assume the constraints factor over vertices, and are applied across the incoming edges to each vertex. That is, for each $v \in V - I$ the parameters $\theta$ restricted to $E_v$ are required to be in a set $\Theta_v \subseteq \mathbf{R}^{E_v}$, and $\Theta = \prod_{v \in V-I} \Theta_v$. (We additionally assume each $\Theta_v$ satisfies constraint qualifications—see Appendix C—and can also alter the factorization requirement to allow more complex network architectures—see Appendix H). If $\Theta = \mathbf{R}^E$, we consider the network to be **unconstrained**. If $\Theta$ is bounded, we consider the network to be **bounded**.

**DLP (Deep Learning Problem)**    Given a loss function $\ell(z, y)$ that is convex in the first argument satisfying $0 \leq \ell(z, y) < \infty$ for all $z \in \mathbb{R}^n$, define $\ell_t(z) = \ell(z, y_t)$ and $L_t(\theta) = \ell_t(c_t(\mathbf{o}, \theta))$. The training problem is to find a $\theta \in \Theta$ that minimizes $L(\theta) = T^{-1} \sum_{t=1}^{T} L_t(\theta)$.

**DLG (Deep Learning Game)**    We define a one-shot simultaneous move game [36, page 9] with infinite action sets (Appendix E); we need to specify the players, action sets, and utility functions.

*Players:* The players consist of a protagonist $p$ for each $v \in V - I$, an antagonist $a$, and a set of self-interested zannis $s_v$, one for each vertex $v \in V$.[2] *Actions:* The protagonist for vertex $v$ chooses a parameter function $\theta_v \in \Theta_v$. The antagonist chooses a set of $T$ vectors and scalars $\{a_t, b_t\}_{t=1}^T$, $a_t \in \mathbb{R}^n$, $b_t \in \mathbb{R}$, such that $a_t^\top z + b_t \leq \ell_t(z)$ for all $z \in \mathbb{R}^n$; that is, the antagonist chooses an affine minorant of the local loss for each training example. Each zanni $s_v$ chooses a set of $2T$ scalars $(q_{vt}, d_{vt})$, $q_{vt} \in \mathbb{R}$, $d_{vt} \in \mathbb{R}$, such that $q_{vt}z + d_{vt} \leq f_v(z)$ for all $z \in \mathbb{R}$; that is, the zanni chooses an affine minorant of its local activation function $f_v$ for each training example. All players make their action choice without knowledge of the other player's choice. *Utilities:* For a joint action $\sigma = (\theta, \{a_t, b_t\}, \{q_{vt}, d_{vt}\})$, the zannis' utilities are defined recursively following the parial order on vertices. First, for each $i \in I$ the utility for zanni $s_i$ on training example $t$ is $U_{it}^s(\sigma) = d_{it} + q_{it}x_{it}$, and for each $v \in V - I$ the utility for zanni $s_v$ on example $t$ is $U_{vt}^s(\sigma) = d_{vt} + q_{vt} \sum_{u:(u,v)\in E} U_{tu}^s(\sigma)\theta(u,v)$. The total utility for each zanni $s_v$ is given by $U_v^s(\sigma) = \sum_{t=1}^T U_{vt}^s(\sigma)$ for $v \in V$. The utility for the antagonist $a$ is then given by $U^a = T^{-1} \sum_{t=1}^T U_t^a$ where $U_t^a(\sigma) = b_t + \sum_{k=1}^n a_{kt}U_{o_k t}^s(\sigma)$. The utility for all protagonists are the same, $U^p(\sigma) = -U^a(\sigma)$. (This representation also allows for an equivalent game where nature selects an example $t$, tells the antagonist and the zannis, and then everyone plays their actions simultaneously.) The next lemma shows how the zannis and the antagonist can be expected to act.

**Lemma 3** *Given a fixed protagonist action $\theta$, there exists a unique joint action for all agents $\sigma = (\theta, \{a_t, b_t\}, \{q_{vt}, d_{vt}\})$ where the zannis and the antagonist are playing best responses to $\sigma$. Moreover, $U^p(\sigma) = -L(\theta)$, $\nabla_\theta U^p(\sigma) = -\nabla L(\theta)$, and given some protagonist at $v \in V - I$, if we hold all other agents' strategies fixed, $U^p(\sigma)$ is an affine function of the strategy of the protagonist at $v$. We define $\sigma$ as the **joint action expansion for $\theta$**.*

There is more detail in the appendix about the joint action expansion. However, the key point is that if the current cost and partial derivatives can be calculated for each parameter, one can construct the affine function for each agent. We will return to this in Section 3.1.

A **KKT point** is a point that satisfies the KKT conditions [15, 19]: roughly, that either it is a **critical point** (where the gradient is zero), or it is a point on the boundary of $\Theta$ where the gradient is pointing out of $\Theta$ "perpendicularly" (see Appendix C). We can now state the main theorem of the paper, showing a one to one relationship between KKT points and Nash equilibria.

**Theorem 4 (DLG Nash Equilibrium)** *The joint action $\sigma = (\theta, \{a_t, b_t\}, \{q_{vt}, d_{vt}\})$ is a Nash equilibrium of the DLG iff it is the joint action expansion for $\theta$ and $\theta$ is a KKT point of the DLP.*

**Corollary 5** *If the network is unbounded, the joint action $\sigma = (\theta, \{a_t, b_t\}, \{q_{vt}, d_{vt}\})$ is a Nash equilibrium of the DLG iff it is the joint action expansion for $\theta$ and $\theta$ is a critical point of the DLP.*

Finally we note that sometimes we need to add constraints between edges incident on different nodes. For example, in a convolutional neural network, one will have edges $e = \{u, v\}$ and $e' = \{u', v'\}$ such that there is a constraint $\theta_e = \theta_{e'}$ (see Appendix H). In game theory, if two agents act simultaneously it is difficult to have one agent's viable actions depend on another agent's action. Therefore, if parameters are constrained in this manner, it is better to have one agent control both. The appendix (beginning with Appendix B) extends our model and theory to handle such parameter tying, which allows us to handle both convolutional networks and non-convex activation functions (Appendix I). Our theory does not apply to non-smooth activation functions, however (e.g. ReLU gates), but these can be approximated arbitrarily closely by differentiable activations.

## 3.1 Learning Algorithms

Characterizing the deep learning problem as a game motivates the consideration of equilibrium finding methods as potential training algorithms. Given the previous reduction to expert algorithms, we will consider the use of the $L_1$ ball constraint $\Theta_v = \{\theta_v : \|\theta_v\|_1 \leq \beta\}$ at each vertex $v$. For deep learning, we have investigated a simple approach by training independent protagonist agents at each vertex against a best response antagonist and best response zannis [14]. In this case, it is possible

| **Algorithm 1** Main Loop | **Algorithm 2** Regret Matching (RM) |
|---|---|
| On round $k$, observe some $x_t$ (or mini batch) <br> Antagonist and zannis choose best responses <br> $\quad$ which ensures $\nabla U_v^p(\theta_v) = -\nabla L(\theta_v^{(k)})$ <br> $g_v^{(k)} \leftarrow \nabla U_v^p(\theta_v)$ <br> Apply update to $r_v^{(k)}, \rho_v^{(k)}$ and $\theta_v^{(k)} \; \forall v \in V$ | $r_v^{(k+1)} \leftarrow r_v^{(k)} + H(\Theta_v)^\top g_v^{(k)} -$ <br> $\qquad\qquad\qquad \rho_v^{(k)\top} H(\Theta_v)^\top g_v^{(k)}$ <br> $\rho_v^{(k+1)} \leftarrow \left(r_v^{(k+1)}\right)_+ / \left(\mathbf{1}^\top \left(r_v^{(k+1)}\right)_+\right)$ <br> $\theta_v^{(k+1)} \leftarrow H(\Theta_v)\rho_v^{(k+1)}$ |

| **Algorithm 3** Exp. Weighted Average (EWA) | **Algorithm 4** Projected SGD |
|---|---|
| $r_v^{(k+1)} \leftarrow r_v^{(k)} + \eta^{(k)} H(\Theta_v)^\top g_v^{(k)}$ <br> $\rho_v^{(k+1)} \leftarrow \exp(r_v^{(k+1)})/(\mathbf{1}^\top \exp(r_v^{(k+1)}))$ <br> $\theta_v^{(k+1)} \leftarrow H(\Theta_v)\rho_v^{(k+1)}$ | $r_v^{(k+1)} \leftarrow r_v^{(k)} + \eta^{(k)} H(\Theta_v)^\top g_v^{(k)}$ <br> $\rho_v^{(k+1)} \leftarrow L_2\_\text{project\_to\_simplex}(r_v^{(k+1)})$ <br> $\theta_v^{(k+1)} \leftarrow H(\Theta_v)\rho_v^{(k+1)}$ |

to devise interesting and novel learning strategies based on the algorithms for learning from expert advice. Since the optimization problem is no longer convex in a local protagonist action $\theta_v$, we do not expect convergence to a joint, globally optimal strategy among protagonists. Nevertheless, one can develop a generic approach for using the game to generate a learning algorithm.

**Algorithm Outline** On each round, nature chooses a random training example (or mini-batch). For each $v \in V$, each protagonist $v$ selects her actions $\theta_v \in \Theta_v$ deterministically. The antagonist and zannis then select their actions, which are best responses to the $\theta_v$ and to each other.[3] The protagonist utilities $U_v^p$ are then calculated. Given the zanni and antagonist choices, $U_v^p$ is affine in the protagonist's action, and also by Lemma 3 for all $e \in E_v$, we have $\frac{\partial L^t}{\partial w_e} = -\frac{\partial U_v^p(\theta_v)}{\partial w_e}$. Each protagonist $v \in V$ then observes their utility and uses this to update their strategy. See Algorithm 1 for the general loop, and Algorithms 2, 3 and 4 for specific updates.

Given the characterization developed previously, we know that a Nash equilibrium will correspond to a critical point in the training problem (which is almost certain to be a local minimum rather than a saddle point [21]). It is interesting to note that the usual process of backpropagating the sampled (sub)gradients corresponds to computing the best response actions for the zannis and the antagonist, which then yields the resulting affine utility for the protagonists.

### 3.2 Experimental Evaluation

We conducted a set of experiments to investigate the plausibility of applying expert algorithms at each vertex in a feedforward neural network. For comparison, we considered current methods for training deep models, including SGD [3], SGD with momentum [33], RMSprop, Adagrad [6], and Adam [17]. Since none of these impose constraints, they technically solve an easier optimization problem, but they are also un-regularized and therefore might exhibit weaker generalization. We tuned the step size parameter for each comparison method on each problem. For the expert algorithms, RM, EWA and PSGD, we found that EWA and PSGD were not competitive, even after tuning their step sizes. For RM, we initially found that it learned too quickly, with the top layers of the model becoming sparse; however, we discovered that RM works remarkably well simply by initializing the cumulative regret vectors $r_v^{(0)}$ with random values drawn from a Gaussian with large standard deviation $\sigma$.

As a sanity check, we first conducted experiments on synthetic combinatorial problems: "parity", defined by $y = x_1 \oplus \cdots \oplus x_m$ and "folded parity", defined by $y = (x_1 \wedge x_2) \oplus \cdots \oplus (x_{m-1} \wedge x_m)$ [27]. Parity cannot be approximated by a single-layer model but is representable with a single hidden layer of linear threshold gates [11], while folded parity is known to be not representable by a (small weights) linear threshold circuit with only a single hidden layer; at least two hidden layers are required [27]. For parity we trained a $m$-$4m$-$1$ architecture, and for folded parity we trained a $m$-$4m$-$4m$-$1$ architecture, both fully connected, $m = 8$. Here we chose the $L_1$ constraint bound to be $\beta = 10$ and the initialization scale as $\sigma = 100$. For the nonlinear activation functions we used a smooth

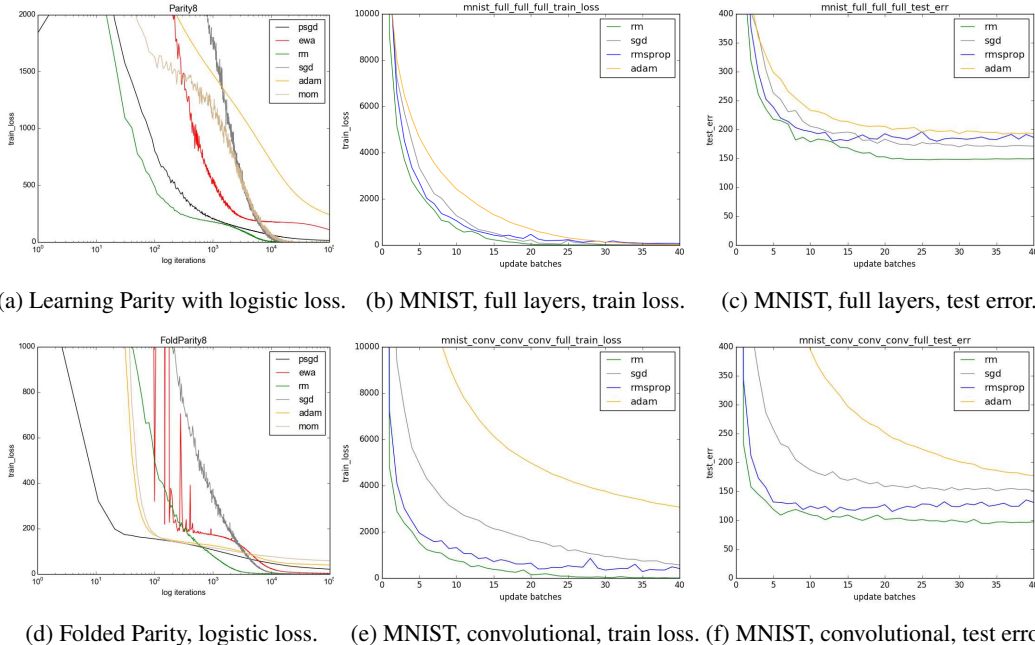

(a) Learning Parity with logistic loss. (b) MNIST, full layers, train loss. (c) MNIST, full layers, test error.

(d) Folded Parity, logistic loss. (e) MNIST, convolutional, train loss. (f) MNIST, convolutional, test error.

Figure 2: Experimental results. (a) Parity, $m$-$4m$-1 architecture, 100 repeats. (d) Folded parity, $m$-$4m$-$4m$-1 architecture, 100 repeats. (b) and (c): MNIST, 784-1024-1024-10 architecture, 10 repeats. (e) and (f): MNIST, $28 \times 28$-$c(5 \times 5, 64)$-$c(5 \times 5, 64)$-$c(5 \times 5, 64)$-10 architecture, 10 repeats.

approximation of the standard ReLU gate $f_v(x) = \tau \log(1 + e^{x/\tau})$ with $\tau = 0.5$. The results shown in Figure 2a and Figure 2d confirm that RM performs competitively, even when producing models with sparsity, top to bottom, of 18% and 13% for parity, and 27%, 19% and 21% for folded parity.

We next conducted a few experiments on MNIST data. The first experiment used a fully connected 784-1024-1024-10 architecture, where RM was run with $\beta = 30$ and initialization scales $(\sigma_1, \sigma_2, \sigma_3) = (50, 200, 50)$. The second experiment was run with a convolutional architecture $28 \times 28$-$c(5 \times 5, 64)$-$c(5 \times 5, 64)$-$c(5 \times 5, 64)$-10 (convolution windows $5 \times 5$ with depth 64), where RM was run with $(\beta_1, \beta_2, \beta_3, \beta_4) = (30, 30, 30, 10)$ and initialization scales $\sigma = 500$. The mini-batch size was 100, and the x-axis in the plots give results after each "update" batch of 600 mini-batches (i.e. one epoch over the training data). The training loss and test loss are shown in Figures 2b, 2c, 2e and 2f, showing the evolution of the training loss and test misclassification errors. We dropped all but SGD, Adam, RMSprop and RM here, since these seemed to dominate the other methods in our experiments. It is surprising that RM can demonstrate convergence rates that are competitive with tuned RMSprop, and even outperforms methods like SGD and Adam that are routinely used in practice. An even more interesting finding is that the solutions found by RM were sparse while achieving lower test misclassification errors than standard deep learning methods. In particular, in the fully connected case, the final solution produced by RM zeroed out 32%, 26% and 63% of the parameter matrices (from the input to the output layer) respectively. For the convolutional case, RM zeroed out 29%, 27%, 28% and 43% of the parameter matrices respectively. Regarding run times, we observed that our Tensorflow implementation of RM was only 7% slower than RMSProp on the convolutional architecture, but 85% slower in the fully connected case.

## 4  Related Work

There are several works that consider using regret minimization to solve offline optimization problems. Once stochastic gradient descent was connected to regret minimization in [4], a series of papers followed [26, 25, 31]. Two popular approaches are currently Adagrad [6] and traditional stochastic gradient descent. The theme of simplifying the loss is very common: it appears in batch gradient and incremental gradient approaches [23] as the majorization-minimization family of algorithms. In the

regret minimization literature, the idea of simplifying the class of losses by choosing a minimizer from a particular family of functions first appeared in [37], and has since been further developed.

By contrast, the history of using games for optimization has a much shorter history. It has been shown that a game between people can be used to solve optimal coloring [16]. There is also a history of using regret minimization in games: of interest is [38] that decomposes a single agent into multiple agents, providing some inspiration for this paper. In the context of deep networks, a paper of interest connects brain processes to prediction markets [1]. However, the closest work appears to be the recent manuscript [2] that also poses the optimization of a deep network as a game. Although the games described there are similar, unlike [2], we focus on differentiable activation functions, and define agents with different information and motivations. Importantly, [2] does not characterize all the Nash equilibria in the game proposed. We discuss these issues in more detail in Appendix J.

## 5  Conclusion

We have investigated a reduction of deep learning to game playing that allowed a bijection between KKT points and Nash equilibria. One of the novel algorithms considered for supervised learning, regret matching, appears to provide a competitive alternative that has the additional benefit of achieving sparsity without unduly sacrificing speed or accuracy. It will be interesting to investigate alternative training heuristics for deep games, and whether similar successes can be achieved on larger deep models or recurrent models.

## Footnotes

*Work performed at Google Brain while on a sabbatical leave from the University of Alberta.

[2] Nomenclature explanation: Protagonists nominally strive toward a common goal, but their actions can interfere with one another. Zannis are traditionally considered servants, but their motivations are not perfectly aligned with the protagonists. The antagonist is diametrically opposed to the protagonists.

[3] Conceptually, each zanni has a copy of the algorithm of each protagonist and an algorithm for selecting a joint action for all antagonists and zannis, and thus do not technically depend upon $\theta_v$. In practice, these multiple copies are unnecessary, and one merely calculates $\theta_v \in \Theta_v$ first.

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
