[Supplementary Material]

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

# A  Proofs for Section 2 (One-layer Case)

We assume that $\ell$ is convex and differentiable in its first argument.

**Fact 6  (OLP Optimality)** *Since each $\ell_t$ in the definition of $L$ is convex and differentiable, $L$ is convex and differentiable, and a necessary and sufficient condition for $\theta^* \in \arg\min_\theta L(\theta)$ is $\nabla L(\theta^*) = 0$ [4, Equation 4.22]*

For the one-layer learning game, OLG, given the simple structure of the utility functions the Nash equilibria are easy to characterize.

**Lemma 7  (OLG Nash Equilibrium)** *The joint action $\sigma = (\theta, \{a_t, b_t\})$ is a Nash equilibrium of the OLG if and only if $\ell_t(\theta x_t) = a_t^\top \theta x_t + b_t$, $a_t = \nabla \ell_t(g)|_{g=\theta x_t}$ (antagonist best response), and $T^{-1} \sum_{t=1}^T a_t x_t^\top = 0$ (protagonist best response).*

**Proof:**  We prove a Nash equilibrium satisfies the two conditions; the converse is similar. The first condition is easiest: since for any $g \in \mathbb{R}^n$, $\ell_t(g) \geq a_t^\top g + b_t$, $\ell_t(\theta x_t) = a_t^\top \theta x_t + b_t$ represents the highest possible utility for the antagonist. Define $z_t = \theta x_t$. However, since $\ell_t$ is convex and differentiable everywhere, there is exactly one affine function that equals $\ell_t$ at a point and is less than or equal everywhere else, namely $h(g) = \nabla \ell_t(z_t)(g - z_t) + \ell_t(z_t)$. Thus, $a_t = \nabla \ell_t(g)|_{g=\theta x_t}$.

Insofar as the protagonist is concerned, given a strategy $a_t$, $b_t$ for the adversary, the protagonist's utility is affine. Specifically, it is $U^p = -T^{-1} \sum_{t=1}^T (a_t)^\top \theta x_t + b_t$. Taking the gradient with respect to $\theta$ yields $-T^{-1} \sum_{t=1}^T a_t x_t^\top$, and setting it to zero guarantees that the protagonist is playing a best response. ∎

**Lemma 8** *For a protagonist action $\theta$, given the best response of the antagonist, $\nabla_\theta U^p = -\nabla L(\theta)$.*

**Proof (of Theorem 1):**  (1) From the conditions stated in Lemma 7 we must have $L(\theta) = T^{-1} \sum_t a_t^\top \theta x_t + b_t$ and $a_t = \nabla \ell_t(g)|_{g=\theta x_t}$ by antagonist best response. By protagonist best response, $T^{-1} \sum_{t=1}^T a_t x_t^\top = 0$, so $T^{-1} \sum_{t=1}^T \nabla \ell_t(g)|_{g=\theta x_t} x_t^\top = 0$. Since $\nabla L(\theta) = T^{-1} \sum_{t=1}^T \nabla \ell_t(g)|_{g=\theta x_t} x_t^\top$, $\nabla L(\theta) = 0$, and therefore by [4, Equation 4.22], $\theta$ is a minimum.

(2) If $\nabla L(\theta^*) = 0$ then $\sum_{t=1}^T \nabla L_t(\theta^*) = 0$. Since $L_t(\theta^*) = \ell_t(\theta^* x_t)$, then $\nabla L_t(\theta^*) = (\nabla \ell_t(g)|_{g=\theta^* x_t}) x_t^\top$. Define $a_t = \nabla \ell_t(g)|_{g=\theta^* x_t}$, and $b_t = \ell_t(\theta^* x_t) - a_t^\top \theta^* x_t$, such that the antagonist is playing a best response to $\theta^*$. Notice that:

$$0 = \nabla L(\theta^*) \tag{2}$$

$$= T^{-1} \sum_{t=1}^T \nabla L_t(\theta^*) \tag{3}$$

$$= T^{-1} \sum_{t=1}^T (\nabla \ell_t(g)|_{g=\theta^* x_t}) x_t^\top \tag{4}$$

$$= T^{-1} \sum_{t=1}^T a_t x_t^\top \tag{5}$$

Thus, $\theta^*$ is also a best response, so $(\theta^*, \{a_t, b_t\})$ is an equilibrium. ∎

For the constrained version of the one-layer neural network, we will temporarily assume that the constraint set $\Theta$ is a polytope. (The set of allowable constraints will be generalized throughout the remainder of the appendix, but linear constraints allow for a simple exposition to start.) Since a polytope is an intersection of a finite set of half-spaces, we can define such a $\Theta$ using a set $J$ of affine functions, where $\theta \in \Theta$ iff for all $j \in J$, $j(\theta) \leq 0$.

To characterize the solutions of the constrained problem, we use the KKT conditions.

**Fact 9  (OCP Optimality)** *Since each $\ell_t$ in the definition of $L$ is convex and differentiable, $L$ is convex and differentiable, so necessary and sufficient conditions for $\theta^* \in \arg\min_{\theta \in \Theta} L(\theta)$ is that there exist*

$\{\mu_j\}_{j \in J}$ such that for all $j \in J$, $\mu_j \geq 0$, $\mu_j j(\theta^*) = 0$, $j(\theta^*) \leq 0$, and: $\sum_{j \in J} \mu_j j(\theta^*) = -\nabla L(\theta^*)$. [4, p. 244].

**Lemma 10 (OCG Nash Equilibrium)** *The joint action $\sigma = (\theta, \{a_t, b_t\})$ is a Nash equilibrium of the OLG if and only if $\ell_t(\theta x_t) = a_t^\top \theta x_t + b_t$, $a_t = \nabla \ell_t(g)|_{g=\theta x_t}$ (antagonist best response), and there exist $\{\mu_j\}_{j \in J}$ such that for all $j \in J$, $\mu_j \geq 0$, $\mu_j j(\theta) = 0$, $j(\theta) \leq 0$, and $\sum_{j \in J} \mu_j j(\theta^*) = -T^{-1} \sum_{t=1}^{T} a_t x_t^\top$ (protagonist best response).*

**Proof:** The antagonist best response proof is nearly identical to the proof of Lemma 7. The protagonist best response leverages that the gradient of $U^p$ with respect to $\theta^*$ is $-T^{-1} \sum_{t=1}^{T} a_t x_t^\top$, and then leverages the KKT conditions for a maximum value. Since $U^p$ is an affine function with respect to $\theta$, a point satisfying the KKT conditions is a global maximum (Fact 9), implying it is a best response for the protagonist. Again, we leave proving the converse as an exercise. ∎

**Proof (of Theorem 2):** (1) Using the same argument as the proof of (1) in Theorem 1, we can argue that $\nabla L(\theta^*) = -\nabla U^p$, as the antagonist best response constraints are the same as before. Using the protagonist best response constraints, we get that there exist $\{\mu_j\}_{j \in J}$ such that for all $j \in J$, $\mu_j \geq 0$, $\mu_j j(\theta^*) = 0$, $j(\theta^*) \leq 0$, and $\sum_{j \in J} \mu_j j(\theta^*) = -T^{-1} \sum_{t=1}^{T} a_t x_t^\top = \nabla U^p$. Thus:

$$\sum_{j \in J} \mu_j j(\theta^*) = -\nabla L(\theta^*) \tag{6}$$

Which are the KKT conditions from Fact 9, so $\theta^*$ is globally optimal.

(2) Assume that $\theta^*$ is an optimal solution. Using the same argument as the proof of (2) in Theorem 1, we construct $a_t$ and $b_t$ in the exact same way, as the antagonist best response property is the same as before, and as easily satisfied. This also implies that $\nabla L(\theta^*) = -\nabla U^p$. Using Fact 9, there exist $\{\mu_j\}_{j \in J}$ such that for all $j \in J$, $\mu_j \geq 0$, $\mu_j j(\theta^*) = 0$, and: $\sum_{j \in J} \mu_j j(\theta^*) = -\nabla L(\theta^*)$. So, $\sum_{j \in J} \mu_j j(\theta^*) = \nabla U^p = -T^{-1} \sum_{t=1}^{T} a_t x_t^\top$. Therefore, the protagonist best response property holds, and $(\theta^*, a_t, b_t)$ is a Nash equilibrium. ∎

## B    Groups of Nodes

To handle two important extensions of feedforward neural networks (e.g. convolutional neural networks (Appendix H) and nonconvex activation functions (Appendix I)), we will need to extend the basic neural network model from the main body of the paper to consider constraints that couple the parameters defined on different edges. For example, in a convolutional neural network, you may have two edges $e = \{u, v\}$ and $e' = \{u', v'\}$ where there is a constraint that $\theta_e = \theta_{e'}$ (see Appendix H). In game theory, if two agents act simultaneously, it is difficult to have one agent's viable actions dependent upon the other agent's action. Thus, if two parameters are jointly constrained, it is best to have one agent control both parameters. Therefore, throughout the remainder of the appendix we will use a generalization of the model described in the body of the paper.

Define a partition $P$ of $V - I$, where for each $\rho \in P$, $E_\rho = \cup_{v \in \rho} E_v$. Also define $\Theta_\rho \subseteq \mathbf{R}^{E_\rho}$, and $\Theta = \prod_{\rho \in P} \Theta_\rho$. An important constraint is that for any $\rho \in P$, and for any $u, v \in \rho$, if $u \leq v$ or $v \leq u$, then $u = v$. We leave the zannis' and the adversaries' action spaces unchanged (i.e., there is still one zanni per node), but each protagonist controls a partition of nodes. Notice that this is a strict generalization of the earlier model, because one could always define the discrete partition where each node is its own partition.

## C    Best Response and KKT Conditions

In this paper, we must analyze partial problems (related to best response in the game) of the form:

**Partial problem at $\rho \in P$:** For an affine function $u : \mathbb{R}^{E_\rho} \to \mathbf{R}$, find $\operatorname{argmax}_{\theta_\rho \in \Theta_\rho} u(\theta_\rho)$.

For each $\rho \in P$, we will define $H_\rho \subseteq \mathbb{R}^{\mathbb{R}^{E_\rho}}$ and $J_\rho \subseteq \mathbb{R}^{\mathbb{R}^{E_\rho}}$ to be finite sets of continuous, differentiable functions. Then, we can define $\Theta_\rho$ to be the set of all $\theta_\rho \in \mathbf{R}^{E_\rho}$ where for all $h \in H_\rho$,

$h(\theta_\rho) = 0$, and for all $j \in J_\rho$, $j(\theta_\rho) \leq 0$. Before we look at the KKT conditions for the partial problem, we define two variations of constraint qualification:

1. **Partial affine constraint qualification:** For all $\rho \in P$, all $h \in H_\rho$ are affine and all $j \in J_\rho$ are affine.

2. **Partial Slater's constraint qualification:** For all $\rho \in P$, all $h \in H_\rho$ are affine and all $j \in J_\rho$ are convex, and there exists a $\theta_\rho \in \Theta_\rho$ where for all $j \in J_\rho$, $j(\theta_\rho) < 0$.

One classic constraint is $\sum_{e \in E_v} |\theta_e| \leq 1$, a bound on the L1 norm of the parameters of a particular vertex. This can be written as a set of linear inequalities (i.e. affine functions in $J_\rho$). Another is $\sum_{e \in E_v} (\theta_e)^2 \leq 1$, a bound on the L2 norm of the parameters of a particular vertex. This can be written as a convex constraint (i.e. $j(\theta_p) = \sum_{e \in E_v} (\theta_e)^2 - 1$).

We will define $\theta_\rho$ to be a **KKT point for a partial problem at $\rho \in P$** if $\theta_\rho \in \Theta_\rho$ and there exists KKT multipliers $\mu_j \geq 0$ and $\lambda_h \in \mathbb{R}$ such that:

$$\nabla u(\theta_\rho) = \sum_{j \in J_\rho} \mu_j \nabla j(\theta_\rho) + \sum_{h \in H_\rho} \lambda_h \nabla h(\theta_\rho) \tag{7}$$

$$\mu_j j(\theta_\rho) = 0 \text{ for all } j \in J_\rho \tag{8}$$

In other words, the gradient points directly out of the feasible $\Theta_\rho$.

**Theorem 11** *[18, 22, 35, 4] Given either the partial affine constraint qualification, or the partial Slater's constraint qualification, any global minimum is a KKT point, and any KKT point is a global minimum.*

Notice that for the partial problem, we have assumed that the utility is an affine function, otherwise a KKT point would not necessarily be a global minimum. We will not make the same assumption for the full problem.

# D Deep Learning and KKT Conditions

Now we will switch and consider the deep learning problem. In the deep learning problem, we want to find a **global minimum** $\theta^* \in \Theta$, such that for all $\theta \in \Theta$, $L(\theta^*) \leq L(\theta)$. This global minimum does not necessarily exist, nor is it necessarily unique. We can also define a distance function over $\mathbb{R}^E$, where for all $\theta, \bar{\theta} \in \Theta$, $d(\theta, \bar{\theta}) = \left( \sum_{e \in E} (\theta(e) - \bar{\theta}(e))^2 \right)^{1/2}$. Define $N \subseteq \Theta$ to be a **neighborhood** of $\theta$ if there exists an $r > 0$ such that for all $\bar{\theta} \in \Theta$, $d(\theta, \bar{\theta}) < r$. $\theta$ is a **local minimum** if there exists a neighborhood $N$ of $\theta$ such that for all $\bar{\theta} \in N$, $L(\theta) \leq L(\bar{\theta})$. Notice that a global minimum is a local minimum.

We will define $\theta \in \mathbb{R}^E$ to be a **KKT point** if $\theta \in \Theta$ and there exists KKT multipliers $\mu_j \geq 0$ and $\lambda_h \in \mathbb{R}$ such that:

$$-\nabla L(\theta) = \sum_{j \in J} \mu_j \nabla j(\theta) + \sum_{h \in H} \lambda_h \nabla h(\theta) \tag{9}$$

$$\mu_j j(\theta) = 0 \text{ for all } j \in J \tag{10}$$

In other words, the opposite of the gradient points directly out of the feasible $\Theta$ (this is a minimization problem, not a maximization problem). The KKT conditions explore properties for a point $\theta \in \Theta$ that are necessary (but not sufficient) for it to be a local minimum of some function $L$. They specify $\Theta$ by giving a set of constraints which must hold for all $\theta \in \Theta$.

For all $\rho \in P$, define $\Pi_\rho : \mathbb{R}^E \to \mathbb{R}^{E_\rho}$ such that for all $\theta \in \mathbf{R}^E$, for all $e \in E_\rho$, $(\Pi_\rho(\theta))_e = \theta_e$. We can define $H = \bigcup_{\rho \in P} \{h \circ \Pi_\rho\}_{h \in H_\rho}$ and $J = \bigcup_{\rho \in P} \{j \circ \Pi_\rho\}_{j \in J_\rho}$. Note that a point $\theta \in \mathbf{R}^E$ is in $\Theta = \prod_{\rho \in P} \Theta_\rho$ if and only if, for all $h \in H$, $h(\theta) = 0$, and for all $j \in J$, $j(\theta) \leq 0$.

1. **Full affine constraint qualification:** For all $\rho \in P$, all $h \in H$ are affine and all $j \in J$ are affine.

2. **Full Slater's Constraint Qualification:** For $H$, $J$, for all $\theta \in \Theta$, all $h \in H$ are affine, all $j \in J$ are convex, and there exists a $\theta \in \Theta$ where for all $j \in J$, $j(\theta) < 0$.

**Theorem 12** *[18, 22, 35]Given the full affine constraint qualification or the full Slater's constraint qualification, any local minimum (and thus, the global minimum) is a KKT point.*

The converse is not true. For example, saddle points can be KKT points as well.

**Lemma 13** *The partial affine constraint qualification implies the full affine constraint qualification.*

**Proof:** For any $\rho \in P$, if $f : \mathbf{R}^{E_\rho} \to \mathbf{R}$ is an affine function, then $f \circ \Pi_\rho$ is an affine function. Thus, for all $\rho \in P$, for all $h \in H_\rho$, $h \circ \Pi_\rho$ is an affine function, so all $h \in H$ are affine. Similiarly, all $j \in J$ are affine. ∎

**Lemma 14** *The partial Slater's Constraint Qualification implies the full Slater's constraint qualification.*

**Proof:** As in the proof of Lemma 13, since $H_\rho$ is a set of affine functions, $H$ is a set of affine functions. For each $\rho \in P$, there exists a $\theta_\rho \in \Theta_\rho$ where for all $j \in J_\rho$, $j(\theta_\rho) < 0$. If we define $\theta \in \mathbb{R}^E$ such that for all $\rho \in P$, $\Pi_\rho(\theta) = \theta_\rho$, then for all $j \in J$, $j(\theta) < 0$. Finally, for every $\rho \in P$, for every $j \in J_\rho$, since $j$ is convex and $\Pi_\rho$ is linear, $j \circ \Pi_\rho$ is convex. ∎

# E   A Simultaneous Move Game

At a high level, in a simultaneous move game[40] there is:

1. a set of players $N$
2. a set (finite or infinite) of actions for each player $\Sigma_i$. A joint action set $\Sigma = \prod_{i \in N} \Sigma_i$.
3. a utility function for each player $u_i : \Sigma \to \mathbb{R}$.

For any $i \in N$, define $\Sigma_{-i} = \prod_{j \in N \setminus i} \Sigma_i$. Given $\mathbf{a} \in \Sigma$, we can write $\sigma_{-i} \in \Sigma_{-i}$ where for all $j \in N \setminus i$, $(\sigma_{-i})_j = \sigma_j$. Thus, for $\sigma_{-i} \in \Sigma_{-i}$ and $\sigma_i \in \Sigma_i$, we can define $\sigma_{-i} \circ \sigma_i \in \Sigma$, where $(\sigma_{-i} \circ \sigma_i)_i = \sigma_i$ and for any $j \in N \setminus i$, $(\sigma_{-i} \circ \sigma_i)_j = (\sigma_{-i})_j$.

A strategy $\sigma_i^* \in \Sigma_i$ is a **best response** to $\sigma_{-i} \in \Sigma_{-i}$ if for all $\sigma_i \in \Sigma_i$, $u_i(\sigma_{-i} \circ \sigma_i^*) \geq u_i(\sigma_{-i} \circ \sigma_i^*)$. A strategy $\sigma_i^* \in \Sigma_i$ is also called a best response to $\mathbf{a} \in \Sigma$ if it is a best response to $\sigma_{-i}$. A joint action $\mathbf{a}^* \in \Sigma$ is a **Nash equilibrium** if for all $i \in N$, $\sigma_i^*$ is a best response to $\sigma_{-i}^*$.

# F   Reasonable Actions, Nash Equilibria

Given a joint action for the deep learning game $\mathbf{a} = (\theta, \{a_t, b_t\}, \{q_{v,t}, d_{v,t}\})$, and some $v \in V$, if $f_v$ is convex and differentiable, define the zanni at $v$ to be **reasonable** for $\mathbf{a}$ if for all $t \in \{1 \ldots T\}$, $q_{v,t} = f'_{v,t}(\sum_{u:(u,v) \in E} c_t(u, \theta)\theta(u, v))$, and $f_v(\sum_{u:(u,v) \in E} c_t(u, \theta)\theta(u, v)) = d_{v,t} + q_{v,t}(\sum_{u:(u,v) \in E} c_t(u, \theta)\theta(u, v))$. In other words, the values and the derivatives of $f_v$ and $d_{v,t} + q_{v,t}x$ match for the activation energies present in the graph.

If the loss $l$ is convex and partially differentiable in the first term, then the adversary is **reasonable** if for all $t \in \{1 \ldots T\}$, $a_t^\top c_t(\mathbf{o}, \theta) + b_t = l_t(c_t(\mathbf{o}, \theta))$ and $a_t = \nabla l_t(z)|_{z=c_t(\mathbf{o}, \theta)}$.

It is straightforward to think about strong induction over a partially ordered finite set.

**Fact 15** *Given a finite set $S$, a partial ordering $\leq$ over $S$, and a set $X \subseteq S$, then if for all $s' \in S$, $\{s' \in S : s' < s\} \subseteq X \Rightarrow s \in X$, then $X = S$.*

Note that in strong induction, the base case is just a case $s \in S$ where $\{s' \in S : s' < s\} = \emptyset$.

**Lemma 16** *Assume that for all $v \in V$, $f_v$ is convex and differentiable. Assume $\leq$ is the partial order generated by the directed acyclic graph in the deep network. For any $v \in V$, given a joint action $\mathbf{a} = (\theta, \{a_t, b_t\}, \{q_{v,t}, d_{v,t}\})$ where for all $u \leq v$, the zanni at $u$ is reasonable for $\mathbf{a}$, then $U_{tv}^s(\mathbf{a}) = c_t(v, \theta)$.*

**Proof:** Define $U \subseteq V$ to be the set of all vertices $u \in V$ where $u \leq v$. Define $R \subseteq U$ to be the set of nodes $v$ where $U_{tv}^s(\mathbf{a}) = c_t(v, \theta)$. We can use the partial order of the graph as a partial order over $U$ to prove recursively that $R = U$.

Then, we can prove by strong recursion on this total order that $U_{tu}^s(\mathbf{a}) = c_t(u, \theta)$ if for all $u' < u$, $U_{tu'}^s(\mathbf{a}) = c_t(u', \theta)$.

1. For any $u \in I$ (i.e. the base case), $U_{tu}^s(\mathbf{a}) = d_{u,t} + q_{u,t}(x_{t,u})$, and since the zanni at $u$ is reasonable, $d_{u,t} + q_{u,t}(x_{t,u}) = f_v(x_{t,u}) = c_t(u, \theta)$.

2. For any $u \in U \backslash I$ (i.e. the inductive case), for all $(u', u) \in E$, $u' < u$, so $U_{tu'}^s(\mathbf{a}) = c_t(u', \theta)$, and thus $U_{tu}^s(\mathbf{a}) = d_{u,t} + q_{u,t}(\sum_{u':(u',u) \in E} c_t(u', \theta)\theta(u', u))$ Since the zanni is $u$ is reasonable, $d_{u,t} + q_{v,t}(\sum_{u':(u',u) \in E} c_t(u', \theta)\theta(u', u)) = f_u(\sum_{u':(u',u) \in E} c_t(u', \theta)\theta(u', u)) = c_t(u, \theta)$.

■

**Lemma 17** *Assume that for all $v \in V$, $f_v$ is convex and differentiable, the loss $l$ is convex and partially differentiable in the first term, and given a joint action $\mathbf{a} = (\theta, \{a_t, b_t\}, \{q_{v,t}, d_{v,t}\})$ where all zannis and the adversary are reasonable, then for any example $t$, $U_t^a(\mathbf{a}) = l_t(c_t(\mathbf{o}, \theta))$.*

**Proof:** The proof is analogous to the proof of Lemma 16.  ■

**Lemma 18** *Assume that for all $v \in V$, $f_v$ is convex and differentiable. Assume $\leq$ is the partial order generated by the directed acyclic graph in the deep network. For any $v \in V$, given a joint action $\mathbf{a} = (\theta, \{a_t, b_t\}, \{q_{v,t}, d_{v,t}\})$ where for all $u \leq v$ (except possibly $v$), the zanni at $u$ is reasonable, then unique best response for the zanni at $v$ is to be reasonable.*

**Proof:** Since the zanni knows the example (or equivalently, chooses a different strategy based on the example), fix a specific example $t$. Define $z = x_{t,v}$ if $v \in I$, or $z = \sum_{u:(u,v) \in E} U_{tu}^s(\mathbf{a})\theta(u, v)$ otherwise. Then, the utility of the zanni at $v$ is $d_{v,t} + q_{v,t}(z)$. First, observe that selecting $q_{t,v} = f_v'(z)$ and $d_{t,v} = f_v(z) - f_v'(z)z$ is a legal strategy for the zanni at $v$: because $f_v$ is convex and differentiable, $f_v(x) \geq f_v'(z)(x - z) + f_v(z)$, so by definition $f_v(x) \geq d_{v,t} + q_{v,t}(x)$. Since for any legal strategy for the zanni at $v$, $f_v(z) \geq d_{v,t} + q_{v,t}(z)$, then this strategy maximizes utility for the zanni at $v$, because $f_v(z) = d_{v,t} + q_{v,t}(z)$. Moreover, since $f_v$ is convex and differentiable, this affine function, which is equal to $f_v$ at $z$ and less than or equal to $f_v$ everywhere else, is unique. Finally, note that if $v \in I$, $z = x_{t,v}$, and if $v \notin I$, from Lemma 16, we know that for all $\{u : (u, v) \in E\}$, $U_{tu}^s(\mathbf{a}) = c_t(u, \theta)$, so $z = \sum_{u:(u,v) \in E} c_t(u, \theta)\theta(u, v)$.  ■

Thus, any time all the zannis are playing a best response, they are reasonable, and vice-versa. To complete the story, we consider the adversary.

**Lemma 19** *Assume that for all $v \in V$, $f_v$ is convex and differentiable, the loss $l$ is convex and partially differentiable in the first term, and given a joint action $\mathbf{a} = (\theta, \{a_t, b_t\}, \{q_{v,t}, d_{v,t}\})$ where all zannis are reasonable, then the unique best response for the adversary is to be reasonable.*

**Proof:** The proof is analogous to Lemma 19.  ■

The above lemmas state that it is easy to determine and reason about the best responses of the adversary and the zanni. Now, we go deeper into the analysis to reason about the protagonist.

**Lemma 20** *Assume that for all $v \in V$, $f_v$ is convex and differentiable, the loss $l$ is convex and partially differentiable in the first term, and given a joint action $\mathbf{a} = (\theta, \{a_t, b_t\}, \{q_{v,t}, d_{v,t}\})$ where all zannis are reasonable, and the adversary is reasonable, and the protagonists play $\theta$, then if $U^p$ is the utility of the protagonists, then:*

$$\nabla_\theta U^p(\mathbf{a}) = -\nabla_\theta L(\theta) \tag{11}$$

**Proof:** First, we break apart $U^p$ into $U_t^p$, where $U_t^p$ is the utility of $p$ conditional on nature selecting example $t$.

$$U_t^p(\mathbf{a}) = -U_t^a(\mathbf{a}) \tag{12}$$

$$= -b_t - \sum_{k=1}^{n} a_{t,k}(U_t^s(o_k)) \tag{13}$$

If we can prove $\nabla_\theta U_t^p(\mathbf{a}) = -\nabla_\theta l_t(c_t(\mathbf{o}, \theta))$, the result follows quickly. Taking the partial derivative above, and relying on the lack of outgoing edges from $o_k$:

$$\frac{\partial U_t^p(\mathbf{a})}{\partial U_t^s(o_k)} = -a_{kt} \tag{14}$$

Since the adversary is reasonable, $a_{kt} = \frac{\partial l_t(c_t(\mathbf{o},\theta))}{\partial c_t(o_k,\theta)}$, and:

$$\frac{\partial U_t^p(\mathbf{a})}{\partial U_t^s(o_k)} = -\frac{\partial l_t(c_t(\mathbf{o},\theta))}{\partial c_t(o_k,\theta)} \tag{15}$$

Define $X \subseteq V$ to be the set of all $v \in V$ such that $\frac{\partial U_t^p(\mathbf{a})}{\partial U_{tv}^s(\mathbf{a})} = -\frac{\partial l_t(\mathbf{o},\theta)}{\partial c_t(v,\theta)}$. We consider the partial order $\leq$ over the vertices $V$ in the deep network generated by the directed acyclic graph of the deep network: however we apply induction on the opposite partial order $\sqsubseteq$. We have shown $O \subseteq X$ above. We recursively show $X \supseteq V \backslash O$ below.

For some $v \in V \backslash O$, we want to show $v \in X$, and we assume that for all $u \in V$ where $u \sqsubset v$, $u \in X$. Notice that:

$$\frac{\partial U_t^p(\mathbf{a})}{\partial U_{tv}^s(\mathbf{a})} = \sum_{u:(v,u)\in E} \theta_{v,u} q_{u,t} \frac{\partial U_t^p(\mathbf{a})}{\partial U_{tu}^s(\mathbf{a})} \tag{16}$$

Since $(v,u) \in E$, $u \in V$, $u \sqsubset v$, and $u \neq v$, so by the inductive hypothesis:

$$\frac{\partial U_t^p(\mathbf{a})}{\partial U_{tv}^s(\mathbf{a})} = -\sum_{u:(v,u)\in E} \theta_{v,u} q_{u,t} \frac{\partial l_t(c_t(\mathbf{o},\theta))}{\partial c_t(u,\theta)} \tag{17}$$

Because $\mathbf{a}$ has a reasonable zanni at all $u \in V$, $q_{u,t} = f'_u\left(\sum_{u':(u',u)\in E} \theta(u',u)c_t(u',\theta)\right)$:

$$\frac{\partial U_t^p(\mathbf{a})}{\partial U_{tv}^s(\mathbf{a})} = -\sum_{u:(v,u)\in E} \theta_{v,u} f'_u\left(\sum_{u':(u',u)\in E} \theta(u',u)c_t(u',\theta)\right) \frac{\partial l_t(c_t(\mathbf{o},\theta))}{\partial c_t(u,\theta)} \tag{18}$$

$$\frac{\partial U_t^p(\mathbf{a})}{\partial U_{tv}^s(\mathbf{a})} = -\frac{\partial l_t(c_t(\mathbf{o},\theta))}{\partial c_t(v,\theta)} \tag{19}$$

Now we know for all $v \in V$, $\frac{\partial U_t^p(\mathbf{a})}{\partial U_t^s(v)} = -\frac{\partial l_t(c_t(\mathbf{o},\theta))}{\partial c_t(v,\theta)}$. We can now consider, for any $(u,v) \in E$, the partial derivative with respect to $\theta(u,v)$:

$$\frac{\partial U_t^p(\mathbf{a})}{\partial \theta(u,v)} = \frac{\partial U_t^p(\mathbf{a})}{\partial U_{tv}^s(\mathbf{a})} q_{v,t} U_{tu}^s(\mathbf{a}) \tag{20}$$

$$\frac{\partial U_t^p(\mathbf{a})}{\partial \theta(u,v)} = -\frac{\partial l_t(c_t(\mathbf{o},\theta))}{\partial c_t(v,\theta)} q_{v,t} U_{tu}^s(\mathbf{a}) \tag{21}$$

Because the zannis are reasonable, $q_{v,t} = f'_v(\sum_{u':(u',v)\in E} \theta(u',v)c_t(u',\theta))$ and $U_{tu}^s = c_t(u,\theta)$:

$$\frac{\partial U_t^p(\mathbf{a})}{\partial \theta(u,v)} = -\frac{\partial l_t(c_t(\mathbf{o},\theta))}{\partial c_t(v,\theta)} f'_v\left(\sum_{u':(u',v)\in E} \theta(u',v)c_t(u',\theta)\right) c_t(u,\theta) \tag{22}$$

Since $c_t(v,\theta) = f_v(\sum_{u':(u',v)\in E} \theta(u',v)c_t(u',\theta))$, $\frac{\partial c_t(v,\theta)}{\partial \theta(u,v)} = f'_v(\sum_{u':(u',v)\in E} \theta(u',v)c_t(u',\theta))c_t(u,\theta)$, hence

$$\frac{\partial U_t^p(\mathbf{a})}{\partial \theta(u,v)} = -\frac{\partial l_t(c_t(\mathbf{o},\theta))}{\partial c_t(v,\theta)} \frac{\partial c_t(v,\theta)}{\partial \theta(u,v)} \tag{23}$$

$$\frac{\partial U_t^p(\mathbf{a})}{\partial \theta(u,v)} = -\frac{\partial l_t(c_t(\mathbf{o},\theta))}{\partial \theta(u,v)} \tag{24}$$

Averaging across examples yields the result. ∎

For the deep network graph, define $P(u, v)$ to be the set of all paths from $u$ to $v$, and for any path $p$, define $|p|$ to be the number of nodes in the path. Thus, for all $p \in P(u, v)$, $p_1 = u$ and $p_{|p|} = v$. The following lemma establishes the partial derivative with respect to $\theta(u, v)$. The key point of the lemma is that if $(u, v), (u', v') \in E_\rho$, then the partial derivative of $\theta(u, v)$ does not depend upon $\theta(u', v')$, and therefore if we restrict ourselves to modifying the weights in $\theta_\rho$, $U^p$ is affine.

**Lemma 21**

$$\frac{\partial U^p(\mathbf{a})}{\partial \theta(u, v)} = -\frac{1}{T} \sum_{t=1}^{T} \sum_{k=1}^{n} \sum_{p \in P(v, o_k)} U^s_{tu}(\mathbf{a}) q_{t, p_{|p|}} a_{kt} \prod_{j=1}^{|p|-1} \theta(p_j, p_{j+1}) q_{t, p_j} \quad (25)$$

**Proof:** Notice that:

$$\frac{\partial U^p(\mathbf{a})}{\partial \theta(u, v)} = \sum_{t=1}^{T} U^s_{tu}(\mathbf{a}) q_{t, v} \frac{\partial U^p(\mathbf{a})}{\partial U^s_{tv}(\mathbf{a})} \quad (26)$$

Thus, the problem can be reduced to proving recursively, starting from $O$:

$$\frac{\partial U^p(\mathbf{a})}{\partial U^s_{tv}(\mathbf{a})} = -\frac{1}{T} \sum_{k=1}^{n} a_{kt} \sum_{p \in P(v, o_k)} \prod_{j=1}^{|p|-1} \theta(p_j, p_{j+1}) q_{t, p_{j+1}} \quad (27)$$

∎

To clarify the decisions of the protagonists, for each $\rho \in P$, define $U^p_{\rho, \mathbf{a}} : \mathbf{R}^{E_\rho} \to \mathbf{R}$ such that $U^p_{\rho, \mathbf{a}}(\theta_\rho)$ is the utility of the protagonist at $\rho$ if she unilaterally deviates from $\mathbf{a}$ to play $\theta_\rho$.

**Lemma 22** $U^p_{\rho, \mathbf{a}}$ *is an affine function.*

**Proof:** Fix a specific $\mathbf{a} = (\theta, \{a_t, b_t\}, \{q_{v,t}, d_{v,t}\})$. We can define $\mathbf{a}|_\rho : \Theta_\rho \to \Sigma$ such that for any $\tilde{\theta} \in \Theta_\rho$, $\mathbf{a}|_\rho(\tilde{\theta})$ is the same as $\mathbf{a}$ except the action of the protagonist at $\rho$ is replaced by $\tilde{\theta}$. So:

$$U^p_{\rho, \mathbf{a}}(\tilde{\theta}) = U^p(\mathbf{a}|_\rho(\tilde{\theta})) \quad (28)$$

This tortured nomenclature allows us to say, for any $(u, v) \in E_\rho$:

$$\frac{\partial U^p_{\rho, \mathbf{a}}(\tilde{\theta})}{\partial \tilde{\theta}_{(u,v)}} = \frac{\partial U^p(\mathbf{a}|_\rho(\tilde{\theta}))}{\partial \tilde{\theta}_{(u,v)}} \quad (29)$$

From Lemma 21

$$\frac{\partial U^p(\mathbf{a}|_\rho(\tilde{\theta}))}{\partial \tilde{\theta}(u, v)} = -\frac{1}{T} \sum_{t=1}^{T} \sum_{k=1}^{n} \sum_{p \in P(v, o_k)} U^s_{tu}(\mathbf{a}|_\rho(\tilde{\theta})) q_{t, p_{|p|}} a_{kt} \prod_{j=1}^{|p|-1} \theta(p_j, p_{j+1}) q_{t, p_j} \quad (30)$$

Thus, the function $U^p_{\rho, \mathbf{a}}$ is differentiable everywhere. Moreover, consider $U^s_{tu}(\mathbf{a}|_\rho(\tilde{\theta}))$. Notice $v \in \rho$. $U^s_{tu}$ is the output of node $u$: thus, since for all $u' \in \rho \backslash \{v\}$, $u' \not\preceq v$, then neither $u$ nor any ancestor is in $\rho$. So, $U^s_{tu}$ is unaffected by changing $\theta_\rho$. More specifically, $U^s_{tu}(\mathbf{a}|_\rho(\tilde{\theta})) = U^s_{tu}(\mathbf{a})$. So:

$$\frac{\partial U^p(\mathbf{a}|_\rho(\tilde{\theta}))}{\partial \tilde{\theta}(u, v)} = -\frac{1}{T} \sum_{t=1}^{T} \sum_{k=1}^{n} \sum_{\rho \in P(\rho, o_k)} U^s_{tu}(\mathbf{a}) q_{t, p_{|p|}} a_{kt} \prod_{j=1}^{|p|-1} \theta(p_j, p_{j+1}) q_{t, p_j} \quad (31)$$

So, the partial derivative is a function only of $\mathbf{a}$, not $\tilde{\theta}$. A function with a constant partial derivative along every coordinate is affine. ∎

We now prove a more general version of Lemma 3.

**Lemma 23** *Given a fixed protagonist action $\theta$, there exists a unique joint action for all agents $\sigma = (\theta, \{a_t, b_t\}, \{q_{vt}, d_{vt}\})$ (the joint action expansion) where the zannis and the antagonist are playing best responses to $\sigma$. Moreover, $U^p(\sigma) = -L(\theta)$, $\nabla_\theta U^p(\sigma) = -\nabla L(\theta)$, and given some protagonist at $\rho \in P$, if we hold all other agents' strategies fixed, $U^p(\sigma)$ is an affine function of the strategy of the protagonist at $p$.*

**Proof:** Most of the insights are in Lemmas 18, 19, 20, and Lemma 22. We know from above that everyone will be reasonable in the joint action expansion. We just have to carefully construct it to prove that it exists and is unique. Consider a parameter $\theta \in \Theta$, and an arbitrary joint action $\mathbf{a}^0 = (\theta, \{a_t, b_t\}, \{q_{v,t}, d_{v,t}\})$. First of all, given the partial ordering $\leq$ over $V$, consider $\sqsubseteq$ to be a linear extension of $\leq$, such that we construct $v_1 \ldots v_{|V|}$, where $v_k \sqsubseteq v_{k+1}$. Define $\mathbf{a}^k$ such that $\mathbf{a}^k$ is equal to $\mathbf{a}^{k-1}$, except that the zanni at $v^k$ plays a best response to $\mathbf{a}^{k-1}$. Finally, $\mathbf{a}^*$ will be equal to $\mathbf{a}^{|V|}$ except that the adversary plays a best response. We prove recursively that each time a best response needs to be taken by a zanni, it exists and is reasonable, by Lemma 18. Thus, all the zannis are reasonable in $\mathbf{a}^{|V|}$, thus a best response for the antagonist exists and is reasonable in Lemma 19. Therefore, there does exist some joint action $\mathbf{a}^*$ when all zannis and the adversary are playing a best response. We can then prove that this is unique, again by using Lemma 18 and Lemma 19 guarantee that the reasonable action is a unique best response, and the reasonable action depends only upon $\theta$.

Now, we have established that the joint action extension exists and is unique. We now want to prove the other properties described in Lemma 23.

Because the adversary is reasonable, by definition, for all $t$ $a_t^\top c_t(\mathbf{o}, \theta) + b_t = l_t(c_t(\mathbf{o}, \theta))$ and $a_t = \nabla l_t(z)|_{z=c_t(\mathbf{o}, \theta)}$. Because the zannis are reasonable, for all $t$, for all $o \in O$, $c_t(o, \theta) = U^s_{ot}(\mathbf{a})$.

Thus, by the definition of the utility of the antagonist:

$$U^a_t(\mathbf{a}) = b_t + \sum_{k=1}^{n} a_{kt} U^s_{o_k t}(\mathbf{a}) \tag{32}$$

$$U^a_t(\mathbf{a}) = b_t + \sum_{k=1}^{n} a_{kt} c_t(o, \theta) \tag{33}$$

$$U^a_t(\mathbf{a}) = l_t(c_t(\mathbf{o}, \theta)) \tag{34}$$

Therefore, averaging over $t$, $U^a(\mathbf{a}) = L(\theta)$. By Lemma 20, $\nabla_\theta U^{prot}(\mathbf{a}) = -\nabla_\theta L(\theta)$. Finally, from Lemma 22, every protagonist faces an affine utility function if she unilaterally deviates. ∎

**Proof (of Lemma 3):** Lemma 3 is a special case of Lemma 23. ∎

**Lemma 24** *Assume that for all $v \in V$, $f_v$ is convex and differentiable, the loss $l$ is convex and partially differentiable in the first term, and given a joint action $\mathbf{a} = (\theta, \{a_t, b_t\}, \{q_{v,t}, d_{v,t}\})$ where all zannis are reasonable, and the adversary is reasonable, then if the joint action $\theta$ for the protagonists is a KKT point, then the protagonists actions are a best response to $\mathbf{a}$, and $\mathbf{a}$ is a Nash equilibrium.*

**Proof:** To prove that this is a Nash equilibrium, we need to show that for each $\rho \in P$, the protagonist at $\rho$ is playing a best response (all zannis and adversaries are reasonable, so they are playing a best response). In other words, we need to show that, if we considered $U^p(\mathbf{a})$, as a function of the values of $\theta$ on $(u, v) \in E_\rho$, then the current $\theta$ in $\mathbf{a}$ is a global maximum. We do this in two steps.

1. We translate the KKT conditions for full problem with $L$ to KKT conditions for a partial problem on $U^p_{\rho, \mathbf{a}}$, the utility function for the protagonist at $v$ deviating.

2. Because $U^p_{\rho, \mathbf{a}}$ is affine, the KKT conditions for a maximum imply a global maximum (see Theorem 11).

As we established in Lemma 20:

$$\nabla_\theta U^p(\mathbf{a}) = -\nabla_\theta L(\theta) \tag{35}$$

Then, the KKT conditions on the loss imply that there exist KKT multipliers $\mu_{j,\rho}$ and $\lambda_{h,\rho}$ such that:

$$-\nabla L(\theta) = \sum_{\rho \in P} \sum_{j \in J_\rho} \mu_{j,\rho} \nabla j(\theta) + \sum_{\rho \in P} \sum_{h \in H_\rho} \lambda_{h,\rho} \nabla h(\theta) \tag{36}$$

$$\mu_{j,\rho} j(\theta) = 0 \text{ for all } \rho \in P, j \in J_\rho \tag{37}$$

Substituting equation 35:

$$\nabla_\theta U^p(\mathbf{a}) = \sum_{\rho \in P} \sum_{j \in J_\rho} \mu_{j,\rho} \nabla j(\theta) + \sum_{\rho \in P} \sum_{h \in H_\rho} \lambda_{h,\rho} \nabla h(\theta) \tag{38}$$

$$\mu_{j,\rho} j(\theta) = 0 \text{ for all } \rho \in P, j \in J_\rho \tag{39}$$

These are the necessary KKT conditions for $\theta$ to be a local maximum. But it is not sufficient. Choose a particular $\rho \in P$. Define $\theta_\rho \in \Theta_\rho$ to be the action of the protagonist at $\rho$ in $\theta$. Now, if we restrict this to the dimensions in $E_\rho$, only the constraints in $J_\rho$ and $H_\rho$ will vary, so:

$$\nabla_{\theta_\rho} U^p(\mathbf{a}) = \sum_{j \in J_\rho} \mu_{j,\rho} \nabla j(\theta) + \sum_{h \in H_\rho} \lambda_{h,\rho} \nabla h(\theta) \tag{40}$$

$$\mu_{j,\rho} j(\theta) = 0 \text{ for all } j \in J_\rho \tag{41}$$

We can replace $U^p(\mathbf{a})$ with $U^p_{\rho,\mathbf{a}}$. For the strategy $\theta_\rho$ that is a part of $\theta$, we get:

$$\nabla_{\theta_\rho} U^p_{\rho,\mathbf{a}}(\theta_\rho) = \sum_{j \in J_\rho} \mu_{j,\rho} \nabla j(\theta_\rho) + \sum_{h \in H_\rho} \lambda_{h,\rho} \nabla h(\theta_\rho) \tag{42}$$

$$\mu_{j,\rho} j(\theta_\rho) = 0 \text{ for all } j \in J_\rho \tag{43}$$

These are the KKT conditions for $\theta_\rho$ to be a local maximum of $U^p_{\rho,\mathbf{a}}$ in $\Theta_\rho$. Therefore, the protagonist at $\rho$ cannot gain by deviating. Now, by Lemma 22, we know $U^p_{\rho,\mathbf{a}}$ is affine, and so if the KKT conditions for a local maximum are satisfied, so are the KKT conditions for a global maximum. Thus, this implies each protagonist cannot unilaterally[4] improve on $\mathbf{a}$, and therefore this is a Nash equilibrium. ∎

**Theorem 25** *Assume that for all $v \in V$, $f_v$ is convex and differentiable, the loss $l$ is convex and partially differentiable in the first term. For every KKT point $\theta \in \Theta$, there is a Nash equilibrium where the joint action of the protagonists is $\theta$, and for every Nash equilibrium where the joint action of the protagonists is $\theta \in \Theta$, $\theta$ is a KKT point.*

**Proof:** To prove that the Nash equilibrium exists, consider the joint action extension of $\theta$. This is a Nash equilibrium by Lemma 24.

To prove the converse, we run the argument of Lemma 24 in reverse. To prove that given a Nash equilibrium $\mathbf{a} = (\theta, \{a_t, b_t\}, \{q_{v,t}, d_{v,t}\})$, $\theta$ is a KKT point, first observe that for any Nash equilibrium, the zannis and adversaries are reasonable (because they are playing best responses). In other words, $\mathbf{a}$ is the joint action extension of $\theta$. Therefore, $\nabla_\theta U^p(\mathbf{a}) = -\nabla_\theta L(\theta)$. Because the equilibrium is an optimal value for the affine function $U^p_{v,\mathbf{a}}$, the KKT conditions must hold for each protagonist. Combining the KKT conditions for each protagonist gives KKT conditions for maximizing $U^p$ over $\Theta$. Since $\nabla_\theta U^p(\mathbf{a}) = -\nabla_\theta L(\theta)$, we can translate the KKT conditions for maximizing $U^p$ into the KKT conditions for minimizing $L(\theta)$. ∎

**Proof (of Theorem 4, DLG Nash Equilibrium):** This is a variant of Theorem 25. ∎

# G Convexity of Antagonist's and Zanni's Strategy Space

This appendix is a side note and thus the notation is mostly disconnected from the rest of the paper. We do not claim that this is original, but it is important to understand whether antagonists can minimize regret.

In order to do online convex optimization, we must have a convex strategy space. Suppose you have a convex, differentiable function $f : \mathbb{R}^n \to \mathbb{R}$. Consider the set $C$ of all $a \in \mathbb{R}^n$, $b \in \mathbb{R}$ such that for all $z \in \mathbb{R}^n$, $a^\top z + b \le f(z)$.

We want to prove $C$ is convex. Due to various technical issues, it is harder than you think.

**Lemma 26** *For any $a \in \mathbb{R}^n$, if there exists a $b \in R$ such that $(a, b) \in C$, then there exists a $v \in \mathbb{R}$ such that for all $c \le v$, $(a, c) \in C$, and for all $c > v$, $(a, c) \notin C$.*

**Proof:** Since $(a, b) \in C$, for all $z$, $f(z) \geq a^\top z + b$. We can define a function $g(z) = f(z) - (a^\top z + b) \geq 0$ is bounded from below, and therefore has a greatest lower bound $q$. $v = b + q$. Thus, for all $z \in \mathbb{R}^n$, $g(z) \geq q$. Therefore, $f(z) \geq a^\top z + b + q = a^\top z + v$ for all $z \in \mathbb{R}^n$.

1. If $c \leq v$, then for all $z \in \mathbb{R}^n$, $f(z) \geq a^\top z + b + q \geq a^\top z + c$.

2. For any $c > v$, then $c - b > q$, and there exists a $z \in \mathbb{R}^n$ where $c - b > g(z)$. For this $z$, $f(z) < a^\top z + c$.

$\blacksquare$

**Lemma 27** *If $(a^1, b^1) \in C$ and $(a^2, b^2) \in C$, then for any $\lambda \in [0, 1]$, there exists some $b^3 \in \mathbb{R}$ such that $(\lambda a^1 + (1 - \lambda)a^2, b^3) \in C$.*

**Proof:** Define $g^1(z) = (a^1)^\top z + b^1$ and $g^2(z) = (a^2)^\top z + b^2$ Consider the function $g(z) = \max(g^1(z), g^2(z))$. By definition, for all $z \in \mathbb{R}^n$, $g(z) \leq f(z)$.

Now, there are three cases for $g(z)$:

1. for all $z \in \mathbb{R}^n$, $g(z) = g^1(z)$. If this is the case, then $a^1 = a^2 = \lambda a^1 + (1 - \lambda)a^2$, and therefore setting $b^3 = b^1$ works.

2. for all $z \in \mathbb{R}^n$, $g(z) = g^2(z)$. Same as above.

3. or there exists $z^1, z^2 \in \mathbb{R}^n$ such that $g(z^1) = g^1(z^1)$ and $g(z^2) = g^2(z^2)$. If this is the case, then there must exist a $z^3$ where $g(z^3) = g^1(z^3) = g^2(z^3)$. Since $g$ is convex, and $a^1$ and $a^2$ are subgradients at $z^3$, then since the set of subgradients at a point is convex, $a^1\lambda + (1 - \lambda)a^2$ is a subgradient at $z^3$. Therefore, set $b^3 = g(z^3) - (a^1\lambda + a^2(1 - \lambda))^\top z^3$. Since the new function is below $g$, it is below $f$.

$\blacksquare$

**Theorem 28** *The set $C$ described above is convex.*

**Proof:** Consider an arbitrary $(a^1, b^1) \in C$ and $(a^2, b^2) \in C$, and $\lambda \in [0, 1]$. For simplicity, define $a^3 = \lambda a^1 + (1 - \lambda)a^2$.

From Lemma 27, there exists a $b^3$ such that $(a^3, b^3) \in C$. Thus, from Lemma 26, there exists a $v$ such that for all $c \leq v$, $(a^3, c) \in C$, and for all $c > v$, $(a^3, c) \notin C$. Thus, if $\lambda b^1 + (1 - \lambda)b^2 \leq v$, we have proven the theorem.

Let us prove this by contradiction: namely, assume $\lambda b^1 + (1 - \lambda)b^2 > v$. Define $\epsilon = \lambda b^1 + (1 - \lambda)b^2 - v$. Since by definition, $\sup_z((a^3)^\top z + v) - f(z) = 0$, then there must exist some $z \in \mathbb{R}^n$ such that $((a^3)^\top z + v) - f(z) \geq -\epsilon/2$.

Now, for this $z$, $a^1 z + b^1 \leq f(z)$, and so $(a^1)^\top z + b^1 \leq ((a^3)^\top z + v) + \epsilon/2$. Similarly, $(a^2)^\top z + b^2 \leq ((a^3)^\top z + v) + \epsilon/2$. Thus, we can combine these to show

$$(\lambda a^1 + (1 - \lambda)a^2)^\top z + \lambda b^1 + (1 - \lambda)b^2 \leq ((a^3)^\top z + v) + \epsilon/2 \tag{44}$$

By the definition of $a^3$:

$$(a^3)^\top z + \lambda b^1 + (1 - \lambda)b^2 \leq ((a^3)^\top z + v) + \epsilon/2 \tag{45}$$

$$\lambda b^1 + (1 - \lambda)b^2 \leq v + \epsilon/2 \tag{46}$$

However, we defined $\epsilon = \lambda b^1 + (1 - \lambda)b^2 - v$, so $v + \epsilon = \lambda b^1 + (1 - \lambda)b^2$, a contradiction. $\blacksquare$

# H Convolutional Neural Networks

Convolutional networks are an extension of the simple feedforward neural network where edge parameters are tied in a particular manner. In particular, if we consider a convolutional layer $l \in L$ that has a width $w_l$, height $h_l$, and depth $d_l$, the vertices within the layer can be indexed by

$I_l = \{1 \ldots w_l\} \times \{1 \ldots h_l\} \times \{1 \ldots d_l\}$; that is, an individual vertex in layer $l$ can be denoted $v_{l,i,j,k}$. The convolution defined at layer $l$ also has a window: for instance, a window of $5 \times 5$ for layer $l$ means that there is an edge between $v_{l-1,i,j,k}$ and $v_{l,i',j',k'}$ if and only if $|i - i'| \leq 2$ and $|j - j'| \leq 2$. Moreover, two edges $(v_{l-1,i,j,k}, v_{l,i',j',k'})$ and $(v_{l-1,i'',j'',k''}, v_{l-2,i''',j''',k'''})$ in $E$ have equal weight if and only if $i - i' = i'' - i'''$, $j - j' = j'' - j'''$, $k = k''$, and $k' = k'''$. All of these constraints are linear equalities and they all occur in the same layer; therefore we can partition the vertices by layer and obtain a valid partitioning, since no vertex is the ancestor of another vertex within the same layer. Moreover, the equality constraints are all valid for the local linear inequality constraint qualification. We can also add an $L_1$ or $L_2$ bound on the weights, either by the terminal vertex of the edge or for the entire layer. Notice that, in practice, instead of having equality constraints between edges, a single copy of the weights is sufficient.

# I   Non-Convex Activation Functions

At first glance, it might appear that the restriction to convex activation functions is too severe, in the sense that it does not include standard (differentiable) activations such as sigmoid and tanh. However, the ability to partition vertices and tie weights with linear equalities, as developed above, allows any activation function that can be expressed as a *difference* of convex functions to still be exactly modeled within our framework. For example, note that the sigmoid, $\sigma(z) = \frac{1}{1+e^{-z}}$, and $\tanh(z) = \frac{e^z - e^{-z}}{e^z + e^{-z}}$ functions can each be written as a difference of differentiable convex functions: For the sigmoid we have $\sigma(z) = \sigma^+(z) - \sigma^-(z)$ for $\sigma^+(z) = \frac{1}{2}(z + \sigma(z) - \log \sigma(z))$ and $\sigma^-(z) = \frac{1}{2}(z - \sigma(z) - \log \sigma(z))$, which are both convex and differentiable. For tanh we have $\tanh(z) = \tau^+(z) - \tau^-(z)$ for $\tau^+(z) = 2\sigma^+(2z) - \frac{1}{2}$ and $\tau^-(z) = 2\sigma^-(2z) + \frac{1}{2}$ which are also both convex and differentiable.

In general, at a node $v$, we can consider any activation function $f_v$ that can be written as a difference of functions, $f_v = f_v^+ - f_v^-$, such that $f_v^+$ and $f_v^-$ are both smooth and convex. In such cases, we can then simulate the contribution of $f_v$ to the circuit computation by adding two sub-nodes, $v^+$ and $v^-$, below $v$, connecting these to $v$ via two new edges $(v^+, v)$ and $(v^-, v)$, and replacing each edge $(u, v)$ by the pair of edges $(u, v^+)$ and $(u, v^-)$. Also, we assign the differentiable convex activation $f_v^+$ to $v^+$, the differentiable convex activation $f_v^-$ to $v^-$, and replace $f_v$ at $v$ with the identity activation $\tilde{f}_v(z) = z$. Then to ensure $f^+$ and $f^-$ receive the same input, we merely add the parameter tying constraints $\theta_{(u,v^+)} = \theta_{(u,v^-)}$ for each pair of corresponding incoming edges $(u, v^+)$ and $(u, v^-)$. To simulate the desired output value, we merely add the constraints that $\theta_{(v^+,v)} = 1$ and $\theta_{(v^-,u)} = -1$ for the parameters on the new edges $(v^+, v)$ and $(v^-, v)$. Denote the modified edge set by $\tilde{E}$. Then we have

$$\tilde{f}_v \left( \theta(v^+, v) f_{v^+} \left( \sum_{u:(u,v^+) \in \tilde{E}} c_t(u,\theta)\theta(u,v^+) \right) + \theta(v^-, v) f_{v^-} \left( \sum_{u:(u,v^-) \in \tilde{E}} c_t(u,\theta)\theta(u,v^-) \right) \right)$$

$$= f_{v^+} \left( \sum_{u:(u,v^+) \in \tilde{E}} c_t(u,\theta)\theta(u,v^+) \right) - f_{v^-} \left( \sum_{u:(u,v^-) \in \tilde{E}} c_t(u,\theta)\theta(u,v^-) \right) \quad (47)$$

$$= f_{v^+} \left( \sum_{u:(u,v) \in E} c_t(u,\theta)\theta(u,v^+) \right) - f_{v^-} \left( \sum_{u:(u,v) \in E} c_t(u,\theta)\theta(u,v^+) \right) \quad (48)$$

$$= f_v \left( \sum_{u:(u,v) \in E} c_t(u,\theta)\theta(u,v^+) \right). \quad (49)$$

Thus, $\theta(u, v^+)$ in the new graph is just like $\theta(u, v)$ in the old one. That is, the new circuit output at node $v$ is the same as the original circuit output at node $v$, but now the neural network only uses differentiable convex activations at each vertex.

Another solution that works for any differentiable function is that, instead of trying to make the zanni maximize the output of the node, have zanni try to "guess" the derivative and the offset. Specifically, given $z$ is the input to the activation function $f_v$ for example $t$, define $q_{v,t}^* = f_v'(z)$, and $d_{v,t}^* = f_v(z) - f_v'(z)z$, and make the utility of the zanni at $v$ to be $\sum_t (q_{v,t} - q_{v,t}^*)^2 + (d_{v,t} - d_{v,t}^*)^2$. As before, reasonable behavior for the zanni is the unique optimal behavior.

# J   Discussion of [2]

The unpublished manuscript [2] presents a variety of interesting and related ideas. In the "gated game" and "CoG game" proposed therein, agents similar to our protagonists are introduced at every vertex. In the gated game, gates are introduced that act as a function of the strategies of the protagonists. In the CoG game, agents (like zannis) are introduced whose actions are a function of the protagonists' actions.

However, the game representations proposed here and in [2] are fundamentally different. The utilities and available information to an agent in a game is crucial in game theory. For instance, Stackelberg games and simultaneous move games are fundamentally different. In a Stackelberg game, one player moves first, and the second observes their movement. In a simultaneous move game, both players must select their strategy independently. For instance, consider a cooperative game, where two players each get a dollar if they both say "heads" or both say "tails", but nothing if they say something different. If one assumes both players move at the same time versus one after the other, these are very different games. Moreover, there is a difference between an agent that is motivated to take an action, versus one that is restricted to play a certain action.

The reason that these distinctions are important is that conventional approaches of regret minimization work in the game developed in this paper but not in [2]: here there is no need to define a new type of regret that is particular to deep networks. Given that a key contribution of this paper is a way to think about optimization problems, whether the concept corresponds to a conventional notion of regret or a new notion of regret is quite important. We leverage the technique in [17] where a game is played with one agent minimizing regret and the other playing a best response. Such a result is not magical: in game theory, there is a huge distinction between having a perfect model of your opponent and prescient knowledge of their actions: the former is still a simultaneous move game, whereas the latter is a Stackelberg game with different equilibria. If the protagonists use randomness, then their behavior cannot be predicted perfectly, and [17] cannot be applied. It is also key that zannis and adversaries observe the example selected by chance: otherwise, they would not be able to model the utilities experienced by the protagonist.

A further distinction is the quality of the connection between solution concepts in the learning problem and the proposed game. in this paper, we show that there is a bijection between Nash equilibria and "KKT points". KKT points include both true local minima as well as some saddle points. Thus, this is a more thorough understanding than the one-way implication in [2] about potential games, that local minima of the potential function are pure Nash equilibria. Since we are viewing the game as a window into the minimization, not being able to account for all Nash equilibria is a limitation. Moreover, it is hard to rectify this issue in [2], since there is no standard equivalent to KKT points for non-differentiable, non-convex functions.

An exact potential game is a game where the utilities are equal. [2] makes references to potential games: "Moreover, simple algorithms such as fictitious play and regret-matching converge to Nash equilibria in potential games," yet the given references have results for two player potential games with a finite number of actions [16, 14]. For the gated game, it is unclear whether these results would extend, especially given the complex nature of the loss functions introduced. It is an open question whether regret minimizers in multiplayer (i.e., more than two) potential games converge to a Nash equilibrium, and whether those results hold for more complex strategy spaces. Moreover, for the purposes of the games in this paper, it is important to understand if convergence results hold for games where only a subset of the agents have utilities that are equal, but one can make strong statements about the behaviors of the other agents.

In this paper, we have focused on deep networks with differentiable activation functions and losses. To deal with issues in non-differentiable activation functions, [2] introduces gated games, but these mean that the games are not in a standard form. The gates are sometimes considered to be dependent upon the agent's behaviors (as in when the game is considered as a potential game) and sometimes the gates are considered to be independent of the agent's behaviors (as when minimizing regret); this is partially justified by GRegret, but not the nuances introduced. There are unsupported statements, such as that minimizing regret guarantees correlated equilibrium (it is actually minimizing internal regret guarantees correlated equilibria, not external regret, and GRegret is clearly more closely associated with GRegret). Thus, one cannot use convergence results about games with a finite number of actions without extending said results to the case of gated games.