[Reviews · NeurIPS 2016]

Reviewer 1

Summary

The paper reformulates learning in NNs as a game, and demonstrates a correspondence between Nash equilibria and KKT points. I enjoyed reading the paper and am extremely sympathetic to its aims. However, I have concerns about: (i) whether the zanni's play a significant role; (ii) whether the experiments evaluate anything meaningful, and (iii) whether the results apply to the neural nets that people actually use. If these are suitably addressed, I will modify my scores accordingly. Detailed comments follow.

Qualitative Assessment

Detailed comments 1. Line 36 introduces h(x) = \phi(\theta x) where \phi is a transfer function, which I guess is intended to model the nonlinear activations in the output layer of the (single-layer) NN. However, neither h nor \phi show up in the remainder of the section. Incorporating \phi breaks Theorems 1 and 2 since the loss \ell(\phi(\theta x), y) will not be a convex function of \theta for most choices of function \phi. I guess the authors meant to say “One linear layer learning games”? 2. Generality of results Regarding Theorems 4 and 5, the ancillary lemmas in the appendix assume that the activation functions f_v are convex and differentiable. This means that the most commonly used functions in practice (ReLU, maxout, max-pooling, sigmoid, and tanh units) are all ruled out along with many others. The results in the paper therefore apply in much less generality than a cursory reading would suggest. The authors should clarify which NNs the results apply to, preferably in the introduction. More broadly, can the results be extended to the neural nets used by practitioners? If not, then the title "Deep learning [x]" is a little misleading. 3. Constraints Minor point: The L1 constraint is disconnected from the rest of the paper. Meaning, the reasons for *specifically* choosing L1 are unclear to me. One could just as easily have chosen, say, L2 and used different OCO algorithms than EWA. Why highlight this particular constraint? Sparsity is mentioned a few times, but sparsity is conceptually orthogonal to the paper’s main contribution. 4. Zannis Line 225 states “the usual process of backpropagating the sampled (sub)gradients [computes] the best response actions for the zannis and the antagonist and the resulting affine utility for the protagonists.” This raises two major questions. First question: Is it necessary to introduce zanni’s into the game in the first place? Since their best responses are given by backpropagation, one could directly construct the loss/utility for each protagonist by suitably backpropagating the loss specified by the antagonist — this recovers the path-sum gradient game introduced by Balduzzi in [2]. Continuing, line 210 states “we investigated […] independent protagonist agents at each vertex against a best response antagonist and best response zannis”. So the experiments don’t use the additional degrees of freedom introduced into the setup by introducing zannis as agents in their own right. Instead, the authors presumably just plug in backprop for the zannis. So, again, do the zannis add anything real to the setup? I’m actually curious how the games run when all players including zannis are “alive” and update their responses using one of Algorithms 2-4. For example, does the game still converge to the same or comparable local optima to when the zanni best responses are computed via backprop? How much does convergence slow down? Second question: What do the experiments evaluate? Line 229 states “to investigate the plausibility of applying independent expert algorithms at each vertex [of] deep learning models”. But applying SGD, RMSProp, Adam etc to backpropagated errors is *exactly* how NNs are trained. Since the zanni’s best responses are backpropagated errors, it seems the only difference between standard NN training and the experiments in the paper is that the authors impose constraints on the weights. However, the effect of the constraints (for example, changing the set's size) is not investigated. And of course the effect of constraints isn’t really what the paper is about. So what do the experiments tell the reader? Please clarify this point. Otherwise, aside from the RM result, the experiments come down to running standard optimizers on an NN on MNIST and observing that they work. As an aside, it is very interesting that Regret Matching (RM) is competitive with RMSProp et al in the experiments and yields sparser solutions. However, this needs to be investigated more thoroughly before it can be considered a robust result. It is also largely orthogonal to the main concerns of the paper. Minor point: Is zanni standard terminology? I’ve never encountered it before and find it quite unpleasant. Please consider alternatives.

Confidence in this Review

3-Expert (read the paper in detail, know the area, quite certain of my opinion)


Reviewer 2

Summary

The paper presents a reduction of supervised learning using game theory ideas that interestingly avoids duality. The authors drive the rationale about the connection between convex learning and two-person zero-sum games in a very clear way describing current pitfalls in learning problems and connecting these problems to finding Nash equilibria. They then present state of the art on-line algorithms for finding Nash equilibria and conduct experimental evaluations to investigate how these methods for perform in solving supervised learning tasks. Surprisingly, one of the simplest methods and that require less parameter tuning (regret minimization) performed consistently better than the rest. The paper's flaws include a failure to introduce the reader about the connection between the problem of training a feedforward neural network and the problem being solved. This is an important bit where readers might loose focus easily. Also, images are of very bad quality, to the point that I did not manage to distinguish the difference between lines.

Qualitative Assessment

I believe the problem could have been better introduced, it was never crystal clear what the problem being addressed is. The paper is interesting and relevant, but the audience could be limited to experts in machine learning (with special interest in deep learning) and game theory. There are some points with almost no background about the choices made and although for an expert in both fields the assumptions made could be catched, this is probably not the general case. The deep learning game formulation is a bit mysterious in my opinion. There are two types of players playing at vertices V-I, that said, there seems that there are |V-I| protagonist players, 1 antagonist and |V| zannis. This is never said explicitly and it is confusing how two players can play at the same vertex. Furthermore, the setting seems like a Stackelberg game solution (or a subgame perfect NE) is more appropriate than a simple Nash equilibria, but the authors do not comment about this.

Confidence in this Review

2-Confident (read it all; understood it all reasonably well)


Reviewer 3

Summary

This paper presents a reduce the supervised learning setup (deep learning) to game playing setup. As well, the author shows the relation between Nash equilibria and KKT points. The author starts by demonstrating the reduction of one layer learning problem to game play between protagonist versus antagonist, and then show the reduction from multi-layer neural networks to game play. Also, they demonstrate that regret matching can achieve competitive training performance.

Qualitative Assessment

The paper is nicely organized such that the reader can follow easily. The theorems are well mathematically explained, but some of the concepts need to more elaboration. For example, the protagonist' action corresponds to the weight in learning problem, how should antagonist's action be thought of as in learning problem? What are the reasoning for some of the design choices. For example, why does antagonist have to choose an affine minorant? and why does utility function be in an affine form? (I understand that mathematically these NE action matches KKT points, but intuitive explanation would be helpful) In the future, how should one approach about designing utility function or antagonist constraints? In the proof of Lemma 7, why is all g \in R^n, l_t(g) <= m^tg+b_t?? should it be l_t(g) >= m^tg+b_t?? The explanation of Lemma 7 is bit sloppy. More detailed explanation would make it clear. In the proof of Lemma 10, "Since U^p is an affine function with respect to theta, a point satisfying the KKT conditions is a global maximum, implying it is a best response for the protagonist" requires proof or citation. Starting from Lemma 13,...,18 and theorems in Deep learning game section assume that f_v(z) are convex. However, most of neural network activation functions are not convex. Isn't f_v(z) being convex too strong assumption? As well, f_v(z) being convex is only mentioned in the supplementary material lemmas, shouldn't this be mentioned in the main paper? In the proof of Lemma 15, - What does f_v being convex has to do with "v=f'(z) and b_{t,v} = f_v (z) - m_{t,v} z being legal? - For all strategy, "f_v(z) >= d_{v,t} + q_{v,t}(z) means strategy maximizes utility" - why? line 676 ~ 680 need some elaboration on why it limits to deterministic one and not probabilistic one. - line 38, Shouldn't "l : R^n x R^n -> R" be "l : R^n x R -> R"? - line 57, U^a(\sigma^a, \tilde{\sigma^a}) should be U^a(\sigma^p, \tilde{\sigma^a}) - line 392, \sum^T_{t=1} m_tx_t^T =0 should be 1/T \sum^T_{t=1} m_tx_t^T =0 - line 410, "Lemma" should be "Lemma 7" - line 522, denominator should be \partial c_t (o_k, \theta) - Equation 19, U^s_{tv}(a) should be U^s_{tu}(a) - line 621 "R" should be C -line 689 "where are the ..." should be "where the ..."

Confidence in this Review

2-Confident (read it all; understood it all reasonably well)


Reviewer 4

Summary

This paper provides a reduction from supervised learning using feed-forward networks to a multiplayer simultaneous move game. The game that results from this reduction is a special case of the general learning from expert advice over a finite number of experts problem. This allows the authors to repurpose the algorithms used for this online learning problem to train a feed forward neural networks. They find that the regret matching algorithm provides competitive performance with the popular variants of stochastic gradient descent used in the deep learning literature.

Qualitative Assessment

I found this paper very interesting: the reduction required a nontrivial derivation and the empirical results are satisfying. Given that there is currently no consensus on the best algorithms to use for optimising deep networks, regret matching seems particularly promising because it doesn't depend on hyper parameters (aside from the initialisation standard deviation). Technical quality. There are two core contributions that this paper makes - it proves a bijection between Nash equilibria in a "Deep Learning Game" and the critical points of the supervised learning problem in feed forward neural networks; and it shows that this allows the use of algorithms from the learning from expert advice literature, one of which provides competitive performance with popular SGD algorithms. The reduction is relatively complex, but presented in a fashion that is relatively clear (and it is particularly useful to build from the simple one-layer case). Given the prevalence of rectified linear units in modern deep learning, it would have been interesting to have a discussion on about the effect of non-differentiable convex functions (my guess is that it makes the analysis more difficult because the best response is no longer unique...), even if this was only an empirical discussion - i.e. does the regret matching algorithm still work well when one uses rectified linear units instead of sigmoid activation functions? I would like to know what the activation function used in the experiments was. Potential impact: I think the practical performance of the regret matching algorithm suggests that this is not just an interesting theoretical exercise, and further investigation of a game theoretic approach to optimising deep networks may prove fruitful. Clarity: The paper is well-written and for the most part free from errors. Some minor corrections: there is a repeated "and" in line 97, and in line 382: the inequality should be reversed(?)

Confidence in this Review

2-Confident (read it all; understood it all reasonably well)